# An ancestral apical brain region contributes to the central complex under the control of *foxQ2* in the beetle *Tribolium*

**Bicheng He[1], Marita Buescher[1], Max Stephen Farnworth[1,2], Frederic Strobl[3], Ernst HK Stelzer[3], Nikolaus DB Koniszewski[1], Dominik Muehlen[1], Gregor Bucher[1]\***

[1]Johann Friedrich Blumenbach Institute of Zoology, GZMB, University of Göttingen, Göttingen, Germany; [2]Göttingen Graduate Center for Molecular Biosciences, Neurosciences and Biophysics, Göttingen, Germany; [3]Buchmann Institute for Molecular Life Sciences (BMLS), Goethe University, Frankfurt, Germany

**Abstract** The genetic control of anterior brain development is highly conserved throughout animals. For instance, a conserved anterior gene regulatory network specifies the ancestral neuroendocrine center of animals and the apical organ of marine organisms. However, its contribution to the brain in non-marine animals has remained elusive. Here, we study the function of the *Tc-foxQ2* forkhead transcription factor, a key regulator of the anterior gene regulatory network of insects. We characterized four distinct types of *Tc-foxQ2* positive neural progenitor cells based on differential co-expression with *Tc-six3/optix*, *Tc-six4*, *Tc-chx/vsx*, *Tc-nkx2.1/scro*, *Tc-ey*, *Tc-rx* and *Tc-fez1*. An enhancer trap line built by genome editing marked *Tc-foxQ2* positive neurons, which projected through the primary brain commissure and later through a subset of commissural fascicles. Eventually, they contributed to the central complex. Strikingly, in *Tc-foxQ2* RNAi knock-down embryos the primary brain commissure did not split and subsequent development of midline brain structures stalled. Our work establishes *foxQ2* as a key regulator of brain midline structures, which distinguish the protocerebrum from segmental ganglia. Unexpectedly, our data suggest that the central complex evolved by integrating neural cells from an ancestral anterior neuroendocrine center.
DOI: https://doi.org/10.7554/eLife.49065.001

**\*For correspondence:**
gbucher1@uni-goettingen.de

**Competing interests:** The authors declare that no competing interests exist.

## Introduction

The brain is among the most complex organs found in animals. During development many different types of neurons are specified to build the macrocircuitry of the central nervous system before the microcircuitry is established. Understanding the genetic and cellular underpinnings of brain development has remained one of the major challenges in developmental biology. Many aspects of neural development are conserved in animals but compared to vertebrates, insects have a strongly reduced number of neural cells and the genes involved are usually present in single copy. This has made insects very useful models to study the genetic control of neural development (*Hartenstein and Stollewerk, 2015*; *Technau et al., 2006*). The insect central nervous system is composed of serially homologous segmental ganglia (*Snodgrass, 1935*; *Weber, 1966*). However, the anterior-most part of the brain, the protocerebrum, is of different origin. It stems from an anterior non-segmental tissue dating back to the last common bilaterian ancestor (*Arendt et al., 2008*; *Rempel, 1975*; *Scholtz and Edgecombe, 2006*; *Snodgrass, 1935*; *Strausfeld, 2012*; *Weber, 1966*). Accordingly, a

number of neural patterning genes are expressed in the anterior brain anlagen but not in the trunk of animals from vertebrates to insects (*Acampora et al., 1998*; *Arendt and Nübler-Jung, 1996*; *Arendt et al., 2008*; *Gehring, 1996*; *Hirth et al., 1995*; *Hirth et al., 2003*; *Lowe et al., 2003*; *Posnien et al., 2011b*; *Quiring et al., 1994*; *Sinigaglia et al., 2013*; *Steinmetz et al., 2010*). Conversely, a number of transcription factors that confer spatial identity to trunk neuroblasts (NBs) are expressed in a quite modified way or not at all in the protocerebral neuroectoderm (*Urbach and Technau, 2003a*; *Urbach and Technau, 2003b*). A number of profound structural differences distinguish the protocerebrum from segmental ganglia. The former contains unique structures like the optic lobes, the mushroom bodies and is marked by a set of midline-spanning neuropils, the central complex (CX) (*El Jundi and Heinze, 2016*; *Pfeiffer and Homberg, 2014*; *Snodgrass, 1935*; *Strausfeld, 2012*; *Weber, 1966*). Further, a process called fascicle switching occurs in the commissure of the brain but not the ventral nerve cord. In that process, some neurites leave their commissural fascicle (de-fasciculation) and join another commissural fascicle (re-fasciculation). Fascicle switching leads to the typical X-shaped chiasmata (decussations) of the central body and is essential for CX development (*Boyan et al., 2008*; *Boyan et al., 2017*).

Recently, a molecular subdivision within the protocerebrum was found where an anterior *optix/six3* positive region distinguishes an ancestral neuroendocrine center of animals from a more posterior *otd/otx* positive region (*Kittelmann et al., 2013*; *Steinmetz et al., 2010*). The components and some of their interactions of the anterior gene regulatory network (aGRN) including *six3* and *foxQ2* are conserved within animals (*Hunnekuhl and Akam, 2014*; *Kitzmann et al., 2017*; *Lowe et al., 2003*; *Marlow et al., 2013*; *Range and Wei, 2016*; *Sinigaglia et al., 2013*; *Wei et al., 2009*; *Yaguchi et al., 2008*; *Yaguchi et al., 2010*). Apart from marking neuroendocrine cells throughout animal clades, this neural region gives rise to the apical organ of marine animals including ciliated cells like the apical tuft (*Dunn et al., 2007*; *Marlow et al., 2013*; *Santagata et al., 2012*; *Sinigaglia et al., 2013*; *Wei et al., 2009*). It was proposed that the anterior brain of bilaterians evolved by the fusion of an ancestral apical brain with an ancestral blastoporal brain located at the opposite pole of the animal (*Tosches and Arendt, 2013*). In this model, the ancestral apical brain contained cells for neuroendocrine control and non-visual photoreception and was patterned by the expression of *six3* and *rx*. Based on recent data, *foxQ2* adds to this list of apical markers. The blastoporal nervous system, in contrast, was located at the opposite pole of the animal, performed a sensory contractile function and was marked by the expression of *nk2.1, Pax6* and other genes. Starting from this ancestral condition, the anterior part of the blastoporal system fused with the apical brain to form an evolutionary 'chimera' forming the anterior brain of extant bilaterians. For instance anterior and tuberal hypothalamus and the retina of vertebrates were proposed to be of chimeric origin (*Tosches and Arendt, 2013*). However, it has remained unclear to which non-neuroendocrine structures the apical region might contribute in arthropods, which do not have an apical organ (*Hunnekuhl and Akam, 2014*; *Marlow et al., 2014*). Further, the identity of the neural parts representing the hypothetical chimeric anterior brain has remained enigmatic in insects.

We have been using the red flour beetle *Tribolium castaneum* as complementary model system for brain development for a number of reasons. First, it represents a more ancestral situation of anterior neuroectoderm development. In *Tribolium* and most other insects including the basal hemimetabolan insects, the brain neuroectoderm derives from a subterminal ventral blastodermal region rather than from the anterior dorsal blastoderm as found in *Drosophila*. This difference in blastodermal geometry results in divergent genetic regulation (*Ansari et al., 2018*; *Fu et al., 2012*; *Kittelmann et al., 2013*; *Nunes da Fonseca et al., 2008*; *Posnien et al., 2010*). Second, any profound functional comparison of brain development and function between *Drosophila* and other taxa requires a quite advanced toolkit. With respect to transgenesis, misexpression, genome editing, efficient genome wide RNAi screening and other tools for analysis of gene function, *Tribolium* is a genetic model insect second only to *Drosophila* (*Berghammer et al., 1999*; *Bucher et al., 2002*; *Dönitz et al., 2018*; *Gilles et al., 2015*; *Schinko et al., 2010*; *Schmitt-Engel et al., 2015*; *Tomoyasu and Denell, 2004*; *Trauner et al., 2009*).

In this work we examined the role of the forkhead transcription factor *Tc-foxQ2* in brain development because this gene is exclusively expressed in the region patterned by the aGRN (*Kitzmann et al., 2017*). Further, it is a highly conserved anterior patterning gene. Orthologs of *foxQ2* are involved in anterior-most specification of neural cells in many clades of bilaterians and in cnidarians (*Darras et al., 2011*; *Fritzenwanker et al., 2014*; *Hunnekuhl and Akam, 2014*;

*Leclère et al., 2016*; *Marlow et al., 2013*; *Range and Wei, 2016*; *Range and Wei, 2016*; *Sinigaglia et al., 2013*; *Wei et al., 2009*; *Yaguchi et al., 2008*; *Yaguchi et al., 2010*; *Yu et al., 2003*). A notable exception are amphibians and mammals, where the gene was lost from the genome (*Mazet et al., 2003*). In arthropods, anterior expression of *foxQ2* orthologs was described in *Drosophila* (*fd102C,* CG11152) and *Strigamia maritima*, a myriapod (*Hunnekuhl and Akam, 2014*; *Lee and Frasch, 2004*) correlating with the location of neuroendocrine cells. Recently, we have shown an upstream role of *Tc-foxQ2* in anterior head epidermis patterning in *Tribolium* where it acts together with *six3/optix* upstream in the aGRN to build the labrum (*Kitzmann et al., 2017*).

Intriguingly, an effect on embryonic CX development was noted as well. The two CX neuropils protocerebral bridge (PB) and central body (CB) form partially during embryogenesis in *Tribolium* and some other holometabolan taxa (*Koniszewski et al., 2016*; *Panov, 1959*; *Wegerhoff and Breidbach, 1992*) while in *Drosophila* the first functional CX neuropil appears during pupation. In the ancestral situation represented for instance by the hemimetabolan insect *Schistocerca gregaria*, the entire brain forms during embryogenesis (*Boyan et al., 2017*; *Koniszewski et al., 2016*). In *Tribolium* larvae, the CB neuropil forms a simple bar crossing the midline. Its position prefigures the one of the adult CB but it still lacks columnar structure and subdivision into upper and lower divisions (fan-shaped and ellipsoid body, respectively) (*Koniszewski et al., 2016*). A modification of the shape of the larval CB was observes after RNAi knock-down of *Tc-foxQ2* but apart from that, functional data on the neural function of *foxQ2* had been missing in any protostome.

We found that *Tc-foxQ2* was continuously expressed in anterior median cell clusters from neural progenitors to postmitotic neurons of the late embryonic, larval and adult brains. Based on co-expression with other protocerebral patterning genes we characterized four different types of *Tc-foxQ2* positive neural progenitor cells. Further, we generated genetic neural imaging lines and found that *Tc-foxQ2* positive neurons project through the primary brain commissure and later into the upper unit of the central body (fan-shaped body). *Tc-foxQ2* RNAi-knock-down embryos failed to develop brain midline structures: The primary brain commissure did form but failed to split into the large number of protocerebral commissures. Consequently, the CB formation was abolished. These results identify *Tc-foxQ2* as one of the key factors of embryonic brain development contributing to the different development of the protocerebrum versus segmental ganglia. Unexpectedly, our results show that cells patterned by the aGRN contribute to the CX, a unique protocerebral brain structure. Apparently, the insect CX evolved by integrating cells from the ancestral anterior-most neuroectoderm, which gives rise to the apical organ including the apical tuft in marine animals.

## Results

### *Tc-foxQ2* marks neural progenitor cells with four different molecular identities

In the embryo, *Tc-foxQ2* is expressed in the anterior neuroectoderm from earliest stages onwards indicating a role in the specification of neuroblasts (*Kitzmann et al., 2017*). Hence, we sought to determine which neural progenitor cells (NPCs) expressed *Tc-foxQ2*. To that end we generated an antibody specific for the Tc-FoxQ2 protein (see Materials and methods and *Figure 1—figure supplement 1*) and used an intronic probe of *Tc-asense* (*Tc-ase*) as marker for NPCs, which could be either NBs, intermediate neural progenitors (INPs) of type II NBs or ganglion mother cells (GMCs) (*Boone and Doe, 2008*; *Bowman et al., 2008*). It should be noted that the *Tc-ase* has been shown to be a marker for NBs in *Tribolium* (*Wheeler et al., 2003*) but that its expression in INPs or GMCs is assumed by analogy from *Drosophila*. In addition, cell size and shape together with large nuclei were used to recognize progenitor cells. See *Figure 1—figure supplement 2* for staging according to *Biffar and Stollewerk (2014)* and for the comparison of the signals of exonic and intronic *Tc-ase* probes.

The first protocerebral NPCs delaminate at NS4. The first Tc-FoxQ2$^+$ NPCs, however, emerge at NS8. Here, about 15 Tc-FoxQ2$^+$/*Tc-ase*$^+$ cells were identified (n = 6; *Figure 1A*; *Supplementary file 1*-table 1) forming a large anterior group (blue and green in *Figure 1A''*), a small median group (gray) and a single lateral cell (orange). These groups correspond to three domains into which the *Tc-foxQ2* expression splits in the neuroectoderm (*Kitzmann et al., 2017*). At NS11, the number had decreased to 9–12 cells expressing both markers (n = 6; *Figure 1B*; *Supplementary file 1*-table 1)

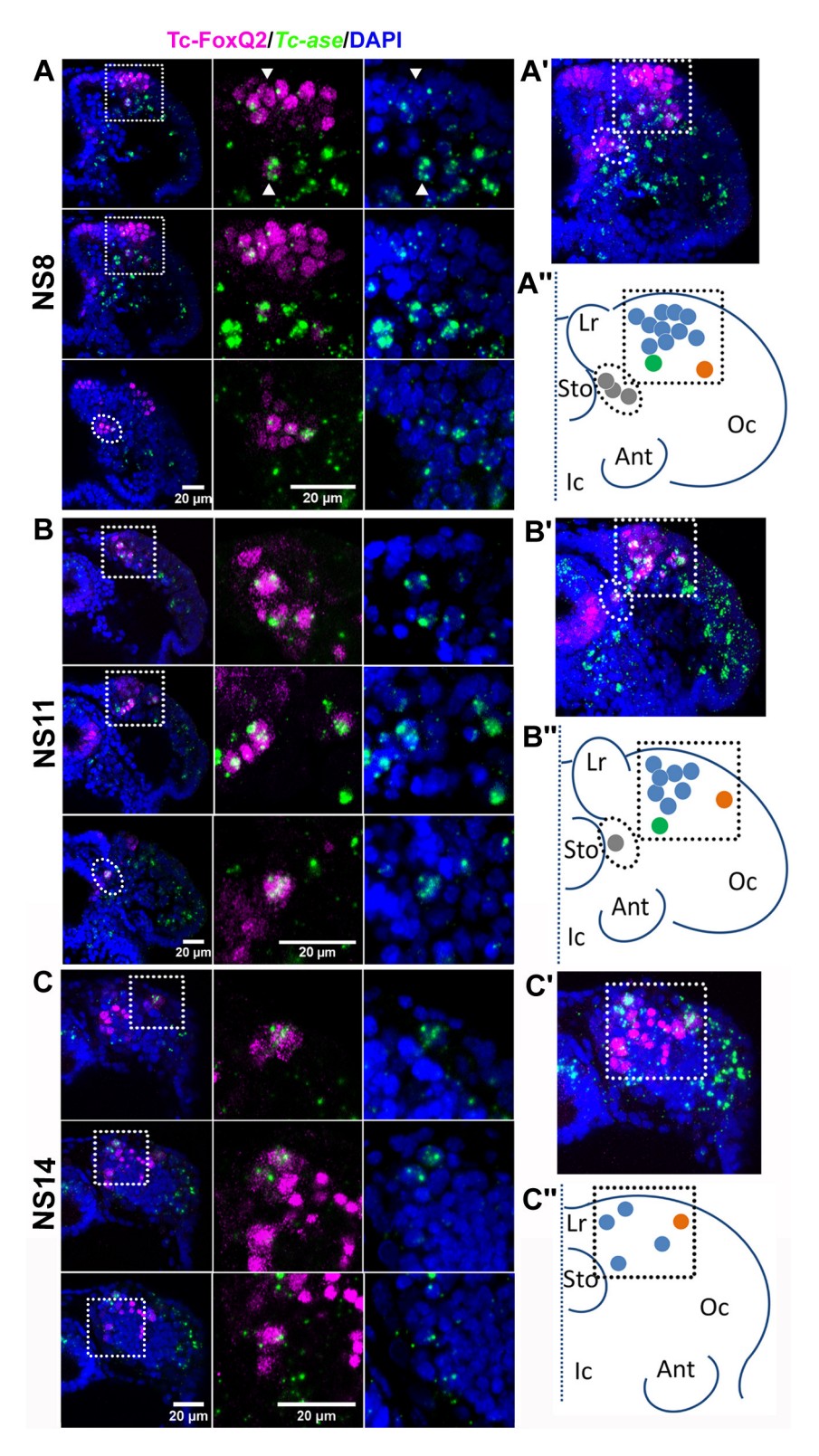

**Figure 1.** Tc-FoxQ2 positive neural progenitor cells. Tc-FoxQ2 protein is visualized by immunohistochemistry at different stages (magenta) while neural progenitor cells (NPCs) are marked by intronic *Tc-asense* whole mount in situ hybridization (green). Nuclei are visualized by DAPI (blue). Optical sections of right halves of stained heads are shown in the left column while respective close-ups are shown in second and third column (see hatched areas in

*Figure 1 continued on next page*

*Figure 1 continued*

left column in (**A**, **B** and **C**). A projection of all optical sections is given in the right column (**A'**, **B'** and **C'**). The schemes represent the outline of right halves of the head lobes of flattened embryos. The dotted line represents the midline. This depiction is comparable to the one previously used for *Drosophila* neuroblast maps (***Urbach and Technau, 2003a***) (**A''**, **B''** and **C''**). (**A–A''**) At NS8 about 15 Tc-FoxQ2 positive NPCs are found (n = 6). By position, three groups are distinguished: A large anterior median group (blue in **A''**) with one neuroblast slightly separated posteriorly (green in **A''**), one single lateral NPC (orange in **A''**) and a group located closely to the midline (gray in **A''**). White arrowheads show two exemplary NPCs. (**B–B''**) At NS11 about 10 Tc-FoxQ2 positive NPCs are observed (n = 6). (**C–C''**) At NS14, the number has decreased to 5–7 cells (n = 6). The single lateral NPC remains distinguishable (orange in **C''**). Lr: labrum; Sto: stomodeum; Oc: ocular region; Ant: antenna; Ic: Intercalary region.

DOI: https://doi.org/10.7554/eLife.49065.002

The following figure supplements are available for figure 1:

**Figure supplement 1.** Generation of a Tc-FoxQ2 antibody.
DOI: https://doi.org/10.7554/eLife.49065.003

**Figure supplement 2.** Developmental staging and comparison of exonic versus intronic *Tc-asense* probes.
DOI: https://doi.org/10.7554/eLife.49065.004

---

while at least 5–7 cells were observed at stage 14 (n = 6; *Figure 1C*; *Supplementary file 1*-table 1). This reduction could have several reasons: *Tc-ase* expression may cease once NBs enter quiescence like in *Drosophila* (*Lai and Doe, 2014*) or *Tc-foxQ2* expression may become repressed in a subset of NPCs. Alternatively, if the anterior group contained type II NBs, the double positive cells would be INPs or GMCs, which based on *Drosophila* knowledge express *asense* while type II NBs themselves do not (*Boone and Doe, 2008*; *Bowman et al., 2008*). Unfortunately, markers for unequivocally distinguishing NBs, INPs and GMCs remain to be established in *Tribolium*.

The identity of NPCs is believed to be determined by unique cocktails of transcription factors (*Skeath, 1999*; *Urbach and Technau, 2003a*). In order to check whether the positional groups were also molecularly distinct we performed co-expression analyses (n = 6 embryos). We used a number of almost exclusive anterior patterning genes because these were likely to contribute to protocerebrum specific patterning (*Posnien et al., 2011a*; *Posnien et al., 2011b*; *Steinmetz et al., 2010*), the co-expression of which we were able to follow until stage NS11 when morphogenetic movements and the increase in cell number made an identification of individually marked cells challenging. We found four different molecular types that correlated with positional differences (*Figure 2*). The first type (No1 in *Figure 2I'*) was the largest group and was located anterior median in the neuroectoderm. This type showed co-expression of *Tc-foxQ2* with *Tc-six3* (*optix*), *Tc-six4* and later *Tc-chx* (*vsx*). We call these cells the *P-fox-am* group of cells (P stands for protocerebral according to *Drosophila* nomenclature [*Younossi-Hartenstein et al., 1996*]). The second type (*P-fox-amp*) was located posteriorly adjacent and was similar to the *P-fox-am* but showed additional expression of *Tc-scro* (*nkx2.1*), *Tc-earmuff* (*fez1*) and in one of the cells also *Tc-eyeless* (*Pax6*) (No2 in *Figure 2I'*). A third type consisted of one lateral NPC, which co-expressed *Tc-six3* with *Tc-earmuff*, *Tc-rx* and *Tc-eyeless* (No3 in *Figure 2I'*). Due to its separate location and molecular distinction, this cell type (*P-fox-l*) could be followed through several stages (orange in *Figure 1*). The fourth type showed only co-expression of *Tc-foxQ2* and *Tc-scro* (No4 in *Figure 2I'*) and was located at a ventral position adjacent to the stomodeum (*P-fox-v*).

We were not able to homologize these NPCs with those of *Drosophila* or *Tenebrio*. This was due to the lack of a comprehensive map of all brain NBs in *Tribolium*, lack of data for most of the respective expression patterns in *Drosophila* NBs and the morphological differences between *Tribolium* and both *Drosophila* and *Tenebrio* (*Urbach and Technau, 2003a*; *Urbach et al., 2003*). However, based on the exclusively pre-antennal expression of *Tc-foxQ2* during embryogenesis (*Kitzmann et al., 2017*), we assign all cells to the protocerebrum. We found no Tc-FoxQ2+ cells in the more posterior parts of the brain or the ventral nerve cord.

Taken together, our analysis showed that *Tc-foxQ2* marks four distinct types of *Tc-ase* positive cells in the early neuroectoderm. Its expression suggests a role in the specification of NPCs in the protocerebral part of the insect brain.

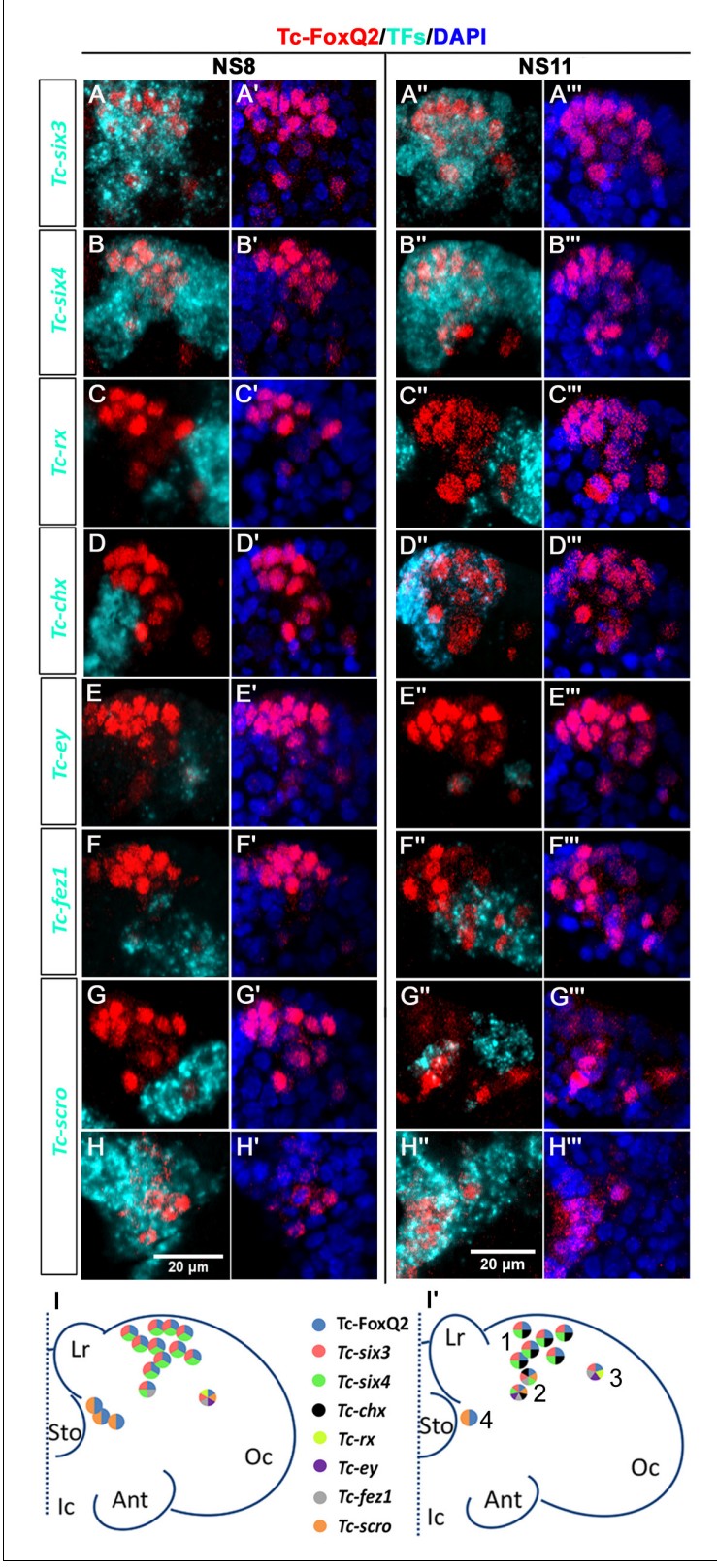

**Figure 2.** Neural progenitor cells co-express transcription factors. Tc-FoxQ2 is visualized by immunohistochemistry (red) while the other transcription factors (TFs) are marked by *fluorescent in situ hybridization* (cyan) at NS8 and NS11. Nuclei are visualized by DAPI (blue). Close-ups of the right halves of embryonic heads are shown (A–H''') and schemes are given at the bottom (I and I'). (A–A''') Co-expression of *Tc-six3* and Tc-FoxQ2. (B–B''') Co-

*Figure 2 continued on next page*

*Figure 2 continued*

expression of *Tc-six4* and Tc-FoxQ2. (C–C''') Co-expression of *Tc-rx* and Tc-FoxQ2. (D–D''') Co-expression of *Tc-chx* and Tc-FoxQ2. (E–E''') Co-expression of *Tc-ey* and Tc-FoxQ2. (F–F''') Co-expression of *Tc-fez1* and Tc-FoxQ2. (G–H''') Co-expression of *Tc-scro* and Tc-FoxQ2. (I, I') Four different identities of NPCs are distinguished based on their position and co-expression. *P-fox-am* (1) is located in the anterior median neuroectoderm. The cells in this group co-express Tc-FoxQ2 with Tc-six3, Tc-six4 and later *Tc-chx*. *P-fox-amp* (2) is located posteriorly adjacent with additional expression of *Tc-scro*, *Tc-fez1* and in one of the cells also *Tc-ey*. *P-fox-l* (3) consists of one lateral cell which co-expresses Tc-FoxQ2 with *Tc-six3*, *Tc-fez1*, *Tc-rx* and *Tc-ey*. *P-fox-v* (4) is located ventrally adjacent to the stomodeum, showing only co-expression of Tc-FoxQ2 and *Tc-scro*. Same abbreviations as in *Figure 1*.

DOI: https://doi.org/10.7554/eLife.49065.005

## Marking of the *Tc-foxQ2* genetic neural lineage by a CRISPR/Cas9 induced enhancer trap

Orthologs of *foxQ2* are involved in anterior-most patterning in animals including development of the apical organ (*Marlow et al., 2014*; *Sinigaglia et al., 2013*; *Yaguchi et al., 2010*). In insects, *foxQ2* is exclusively expressed in protocerebral tissue and a function in protocerebrum-specific neuropils has been suggested (*Kitzmann et al., 2017*). However, *foxQ2*[+] neurons had not been marked to follow their projections in any insect. Unfortunately, *foxQ2* enhancer trap lines were neither available for *Tribolium* nor *Drosophila*. Hence, we used CRISPR/Cas9 genome editing (*Gilles et al., 2015*) for a non-homologous end joining strategy to generate an enhancer trap in the *Tc-foxQ2* locus that drives EGFP (see Materials and methods; *Supplementary file 1*-tables 2–4; *Figure 3—figure supplement 1*, (*Farnworth et al., 2019*). By double immunohistochemistry we confirmed that EGFP and Tc-FoxQ2 protein expression correlated quite well throughout embryogenesis (*Figure 3*). The observed differences appeared to be mainly due to different dynamics of maturation and degradation of the two proteins but we cannot exclude that cells are single positive for either EGFP or Tc-FoxQ2. We called this line *foxQ2-5'-line* and used it for all subsequent analyses of *Tc-foxQ2+* neurons.

During embryogenesis from NS3 to NS13, the Tc-FoxQ2 antibody and EGFP stainings closely matched the in situ hybridization patterns (*Kitzmann et al., 2017*). Essentially, Tc-FoxQ2 was initially expressed in two bilateral anterior-median expression domains (*Figure 3A*). Later, these resolved into a stomodeal (asterisk in *Figure 3B' and F'*) and an anterior domain, which was further subdivided into a median (white arrowhead) and lateral domain (open arrowhead). The *P-fox-v* NPCs were located in the stomodeal domain while the other types emerged from within the anterior domains. Towards the end of embryogenesis (NS15) two clusters of cells reminiscent of neural lineages were observed: A large anterior median group (white arrowhead in *Figure 3F'*) and a smaller lateral group (open arrowhead). In addition, scattered cells were observed more posteriorly in the brain and strong staining in the stomodeum persisted (asterisk).

## Three Tc-FoxQ2 positive cell clusters contribute to brain midline structures including the central body

We characterized the contribution of *Tc-foxQ2*[+] cells to the embryonic brain. We found no *Tc-foxQ2*[+] glia based on immunohistochemistry in our transgenic glia reporter line *glia-blue* (*Koniszewski et al., 2016*) (not shown). In order to study the development of the projection patterns of *Tc-foxQ2*[+] neurons we performed double-immunohistochemistry visualizing the EGFP derived from the *foxQ2-5'-line* combined with ß-acetylated tubulin (acTub), which marks axonal projections (*Piperno and Fuller, 1985*). We detected at least three clusters of *Tc-foxQ2*[+] cells with properties of neural lineages (*Figure 4*, please find entire stacks and videos on figshare project 62939).

At stage NS13, the first brain commissure became visible in the acTub staining (white arrowhead in *Figure 4A'*). One large continuous cluster of about 89 *Tc-foxQ2*[+] cells was situated around this primary commissure and was located in the anterior median part of the forming brain (n = 5; *Figure 4A*; *Supplementary file 1*-table 5). About 12 marked cells had the nuclear morphology of NPCs (n = 5; *Figure 4A*; *Supplementary file 1*-table 5). We termed this group of adjacent cells the *anterior-median-foxQ2-cluster*. Based on the number of observed NPCs within the cluster and the number of projections that emerge from it we assume that it could be composed of several neural lineages. During subsequent development, these cells stayed together but along with general

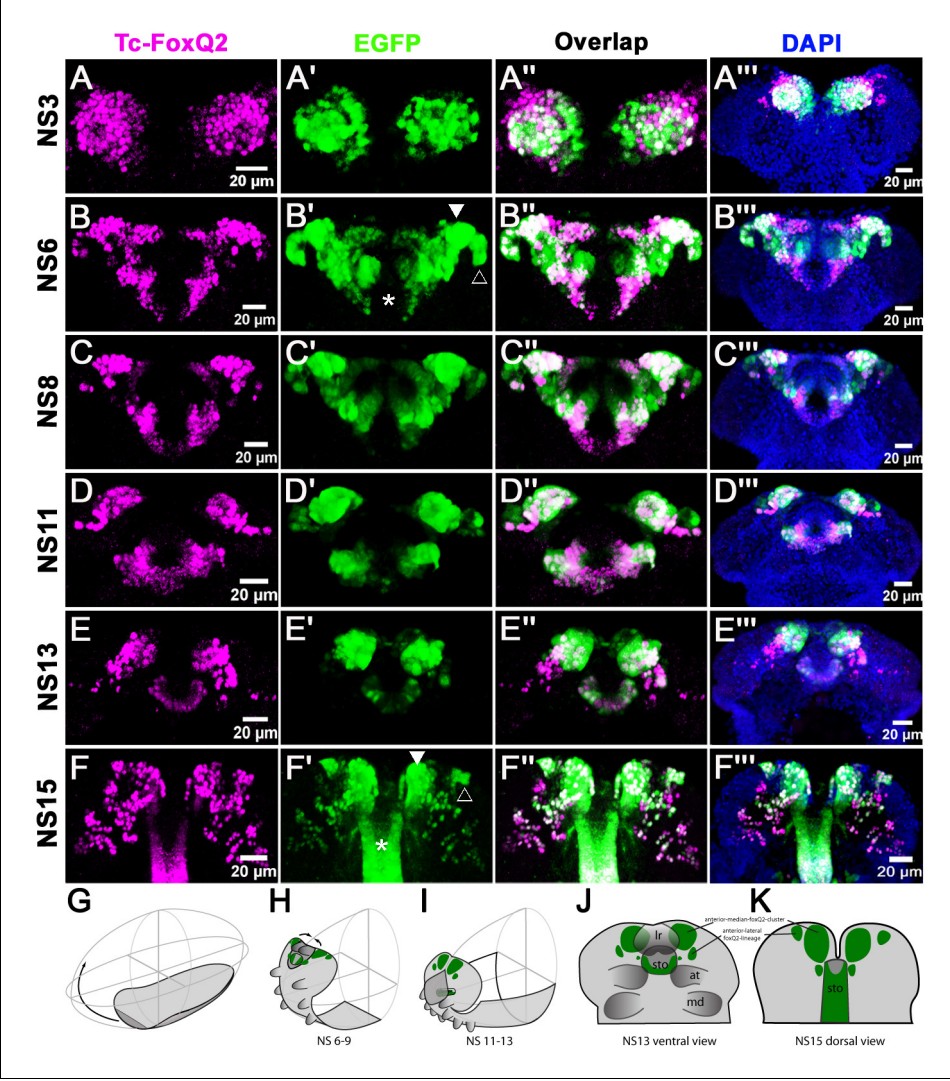

**Figure 3.** *Tc-foxQ2* positive cells marked by antibody and the *foxQ2-5'-line*. The expression of EGFP (green) derived from the *foxQ2-5'-line* and Tc-FoxQ2 protein (magenta) correlate closely throughout embryogenesis. The morphology of the anterior neuroectoderm is visualized with DAPI staining (blue, right column). Shown are heads of embryos dissected out of the egg and flattened to reveal the staining within the neuroectoderm (A'''–C''') and the developing brain (D'''–F'''). Some differences between EGFP and Tc-FoxQ2 expression are observed, which may be due to either different dynamics of maturation and degradation of these proteins or to divergence of the enhancer trap signal from the endogenous expression. (A–A''') At NS3, *Tc-foxQ2* expression shows two bilateral domains within the anterior median region. (B–F''') Later, the expression domains split into a stomodeal (asterisk), a median (white arrowhead) and lateral domain (open arrowhead). At NS15, two clusters of cells are observed: The large *anterior-median-foxQ2-cluster* (white arrowhead in F') and a smaller *anterior-lateral-foxQ2-lineage* (open arrowhead in F'). (G–I) The general movements of the head tissue areshown from the germ rudiment (G) to an elongating (H) and a retracting stage (I). The approximate positions of the *Tc-foxQ2* marked cells underlying the head epidermis are shown. (J,K) Flat preparations of heads of stage NS13 (J) and NS15 (K) are shown with the approximate positions of the underlying *Tc-foxQ2* marked cells shown in green. (G–I) are redrawn from *Posnien and Bucher (2010)*.

DOI: https://doi.org/10.7554/eLife.49065.006

The following figure supplement is available for figure 3:

**Figure supplement 1.** Characterization of the *foxQ2-5'-line*.
DOI: https://doi.org/10.7554/eLife.49065.007

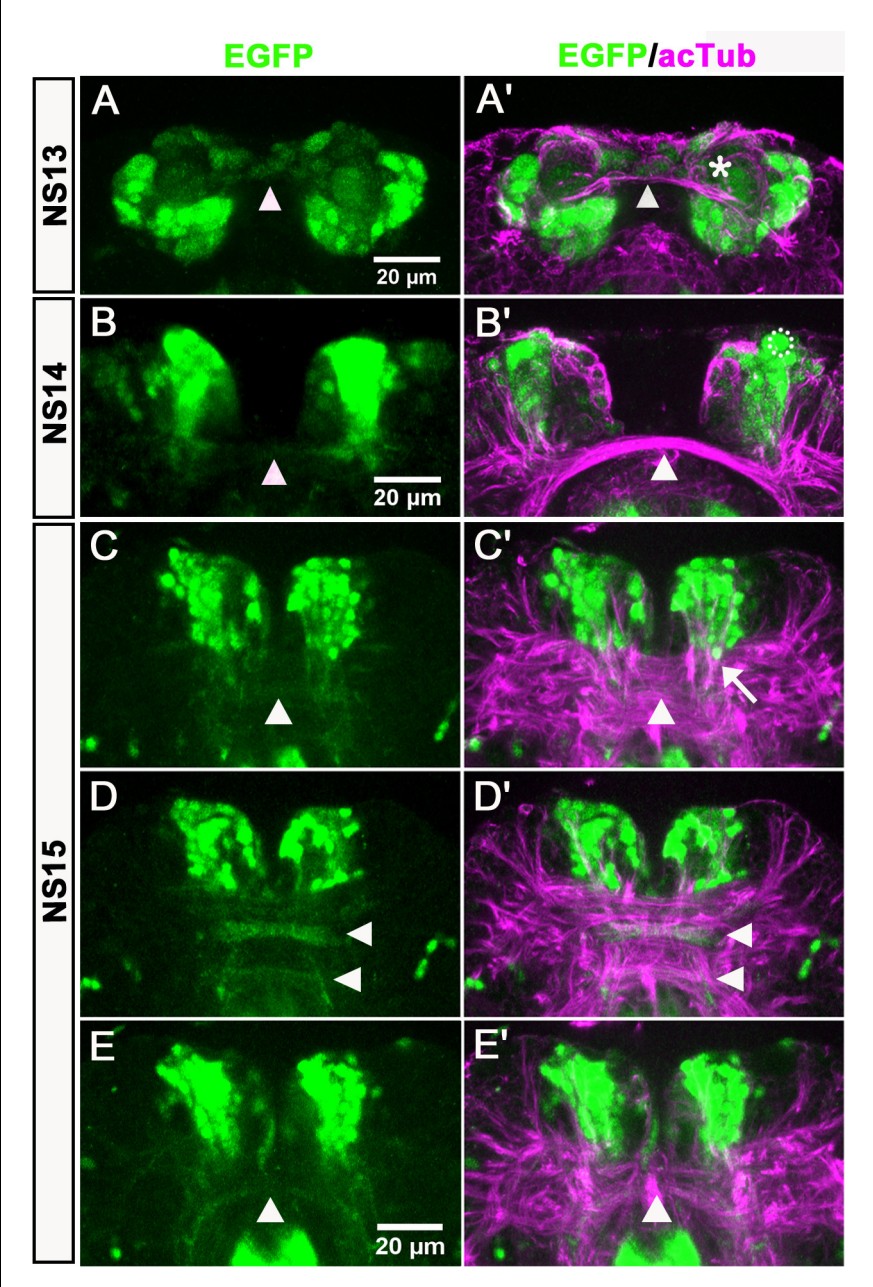

**Figure 4.** *Anterior-median-foxQ2-cluster* contributes to the central brain primordium. Double-immunohistochemistry visualizes the EGFP (green) derived from the *foxQ2-5'-line* and acTub (magenta) which marks axonal projections – neuraxis anterior is up. (**A–A'**) At NS13, the first brain commissure marked by acTub appears (white arrowhead in A'). The cell bodies of the *anterior-median-foxQ2-cluster* are located around this commissure but do not yet project into it. A few weakly stained cells closely attached to the commissure are not Tc-FoxQ2 protein positive (asterisk in A'). (**B–B'**) At NS14, projections within the brain commissure become visible but have not yet reached the midline (white arrowhead in B). One Tc-FoxQ2 positive NPC is recognized by its morphology and position (hatched circle in B'). (**B**) and (**B'**) are not the same embryo but from the same developmental stage. (**C–E**) At NS15, at least three brain commissures are marked by the *anterior-median-foxQ2-cluster*: One in the circumesophageal commissure (white arrowhead in E), and two commissures within the central brain primordium (white arrowheads in D). The *anterior-median-foxQ2-cluster* produces more cells at this stage. (**C'–E'**) acTub marked brain commissures expand into many fascicles and increase in size. 6–7 axon bundles emanating from the *anterior-median-foxQ2-cluster* separately join this midline brain primordium (one of them marked by an arrow in **C'**).

*Figure 4 continued on next page*

*Figure 4 continued*

DOI: https://doi.org/10.7554/eLife.49065.008

The following figure supplement is available for figure 4:

**Figure supplement 1.** *Tc-foxQ2* cell clusters are surrounded by glial sheets.

DOI: https://doi.org/10.7554/eLife.49065.009

morphogenetic movements they approached each other towards the midline (compare distance in *Figure 4B* with E; see *Videos 1* and *2*).

At NS14 the cell number in the *anterior-median-foxQ2-cluster* had increased to about 150 (n = 5; *Figure 4B*; *Supplementary file 1*-table 5) and 14 Tc-FoxQ2$^+$ NPCs were still discernable (e.g. hatched circle in *Figure 4B'*). At that stage, the first EGFP positive projections became visible. They projected towards the brain commissure and joined it. However, at that stage, the EGFP$^+$ projections had not yet reached the midline of the commissure (white arrowhead in *Figure 4B*).

By NS15 the cell number of the *anterior-median-foxQ2-cluster* had increased to approximately 210–240 cells (n = 5; *Figure 4C–E*; *Supplementary file 1*-table 5) but NPCs were no longer distinguishable by morphological means. The brain commissure had split and expanded significantly by additional projections from other lineages (*Figure 4C'–E'*). Likewise, the projections of the *anterior-median-foxQ2-cluster* became more complex. About 6–7 axon bundles emanated from that cluster to separately join the central brain primordium (arrow in *Figure 4C'*) to cross the midline. EGFP signal distinguished at least three major sites of midline structures with *Tc-foxQ2*$^+$ contribution: One in the half-ring-like circumesophageal commissure (white arrowhead in *Figure 4E*), and two separate fascicles within the central brain primordium (white arrowheads in *Figure 4D* and *Figure 4D'*).

In order to assign the *anterior-median-foxQ2-cluster* to characterized NPCs we traced back the EGFP signal from NS15 to the embryonic neuroectoderm. Based on continuous expression from NS3 to NS15 and in vivo imaging data on its development (*Videos 1* and *2*) we suggest that it is derived from *P-fox-am* type of NPCs. By crossing with the transgenic glia marker line *glia-red*

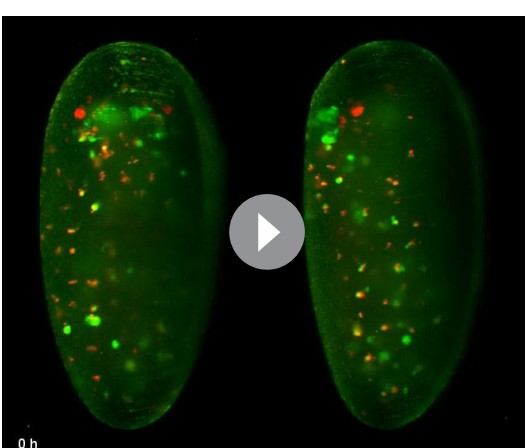

**Video 1.** Shown are a dorsal (left) and lateral (right) view on a developing embryo with *Tc-foxQ2* positive cells marked in green and glial cells marked in red in 10X magnification. Anterior is up. The development is shown over the time of 80 hr. From 0–20 hr the head lobes and the brain anlagen approach each other at the midline. Subsequently, the labrum is seen to elongate towards anterior until 60 hr. Finally, the head undergoes an overall change of position by a rotation towards the ventral side.

DOI: https://doi.org/10.7554/eLife.49065.010

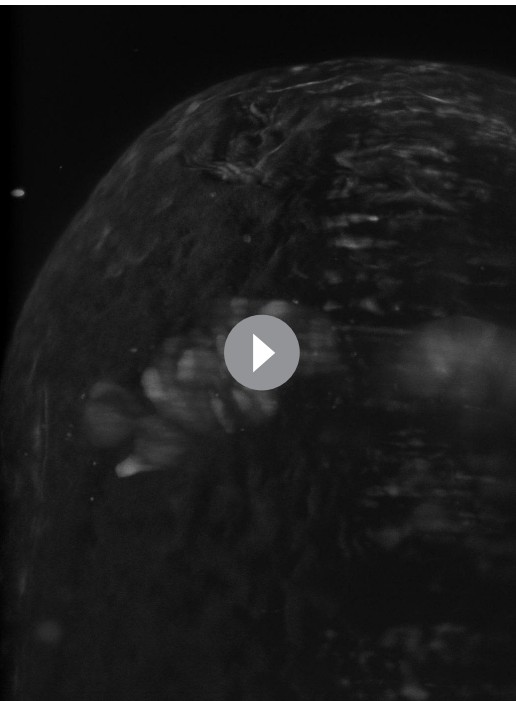

**Video 2.** Shown is the dorsal view on a developing embryo with *Tc-foxQ2* positive cells marked by EGFP at higher magnification (40X). The movements towards the midline of the marked cell clusters are shown.

DOI: https://doi.org/10.7554/eLife.49065.011

(*Koniszewski et al., 2016*) we found that most cells of the cluster were surrounded by one glial sheet indicating that they formed one large neural lineage (white arrowhead in *Figure 4—figure supplement 1*). A second glia sheet appeared to surround the medial-most cells marked with lower levels of EGFP (open arrowhead in *Figure 4—figure supplement 1*).

A second group of EGFP positive cells emerged laterally to that cluster after NS13 (*anterior-lateral-foxQ2-lineage*). At NS14 one NPC and a very small group of cells (6-8) were marked by EGFP (n = 4; orange arrowhead in *Figure 5A*; *Supplementary file 1*-table 6). At NS15 approximately 15–20 cells were marked (n = 4; *Supplementary file 1*-table 6). They formed a column-like cluster lateral to the *anterior-median-foxQ2-cluster* (orange arrowheads in *Figure 5B,D*) and their projections joined the established *Tc-foxQ2* positive neurites at a lateral position (white arrows in *Figure 5B,D*). Because this cluster contained only one NB and all cells were within one glia sheath (orange arrowhead in *Figure 4—figure supplement 1*) we hypothesize that they form one neural lineage. By tracing back this lineage to the embryonic ectoderm we found that it most likely derived from the *P-fox-l* NPC (*Figure 2I'*).

The lineages of anterior median NBs are known to contribute to the CX in other insects (*Boone and Doe, 2008*; *Bowman et al., 2008*; *Boyan et al., 2010*; *Walsh and Doe, 2017*). We showed that some NPCs are *Tc-foxQ2* positive and after RNAi-knock-down of *Tc-foxQ2*, the CX is disturbed (*Kitzmann et al., 2017*). Therefore, we hypothesized that the *Tc-foxQ2* might mark cells contributing to the CX. In order to test this we analyzed larval (L5) and adult brains, where the CB is marked by a glia sheet (white arrowheads in *Figure 5C', F*). Indeed, EGFP marked projections contributed to the upper division (fan shaped body) of the CB (white arrowheads in *Figure 5C,E*) while another fascicle crossed the midline directly dorsal of the CB (dorsal with respect to neuraxis; open arrowhead in *Figure 5E'*).

In summary, we found that *Tc-foxQ2* positive cells formed three cell clusters in the protocerebrum that projected through the early brain commissure. Later, they marked specific subsets of midline fascicles of the developing central brain and eventually contributed to the upper unit of the CB and to subsets of brain commissures.

Some additional *Tc-foxQ2*⁺ cells merit mentioning although they were not neural or did not contribute to the central brain. Firstly, cells of the stomodeum except for the dorsal roof were marked at all stages (*Figure 3* and white arrowheads in *Figure 5—figure supplement 1*). The domain appeared to be continuous with an expression in the lateral parts of the labrum from stage NS14 onwards (open arrowheads in *Figure 5—figure supplement 1*). Secondly, several *Tc-foxQ2*⁺ cells lateral to the stomodeum were observed at NS13 (open arrowhead in *Figure 5—figure supplement 2A*). From NS14 onwards they had expanded to a group of cells adjacent to the stomodeum, which sent projections into the posterior circumesophageal commissure (open arrowhead in *Figure 5—figure supplement 2B*). Thirdly, about 12 weakly stained cells were closely attached to the developing commissure at the midline (n = 5; star in *Figure 4A'*). These cells were Tc-FoxQ2 protein negative at NS13 but could have retained EGFP signal from median cells marked at a previous stage (e.g. dorsal to the stomodeum at NS6; see *Figure 3B'*).

## In vivo imaging reveals complex morphogenetic movements of the developing brain

In order to confirm our view on the morphogenetic behavior of the marked cell clusters we used light-sheet based fluorescence microscopy for in vivo imaging (*Strobl et al., 2015*). We imaged a cross of the *foxQ2-5'-line* with the *AGOC #6* reporter, which marks glia cells via the 3XP3 promoter (*Koniszewski et al., 2016*; *Strobl et al., 2018*). Three prominent morphogenetic movements were revealed: Initially, both brain and stomodeum EGFP signals started out in close vicinity at the dorsal side (*Figure 6A* white arrowhead and white arrow, respectively, *Video 1*). Shortly later, the stomodeum became elongated and bent away towards the ventral side such that the initially adjacent expression domains came to lie on opposite sides of the brain (*Figure 6B–C*). At the same time a second movement was observed, where the bilateral *Tc-foxQ2*⁺ cell clusters converged from lateral positions towards the midline until they made contact at the medial brain (white arrowheads in *Figure 6E–J*, *Video 2*). The third movement consisted of an overall ventral bending of the head and brain where the relative positions of the expression domains remained similar (compare white lines in *Figure 6C and D*). These movements reflected the movements described by the 'bend and zipper' model of head development (*Figure 6K–N*) (*Posnien and Bucher, 2010*; *Posnien et al., 2010*).

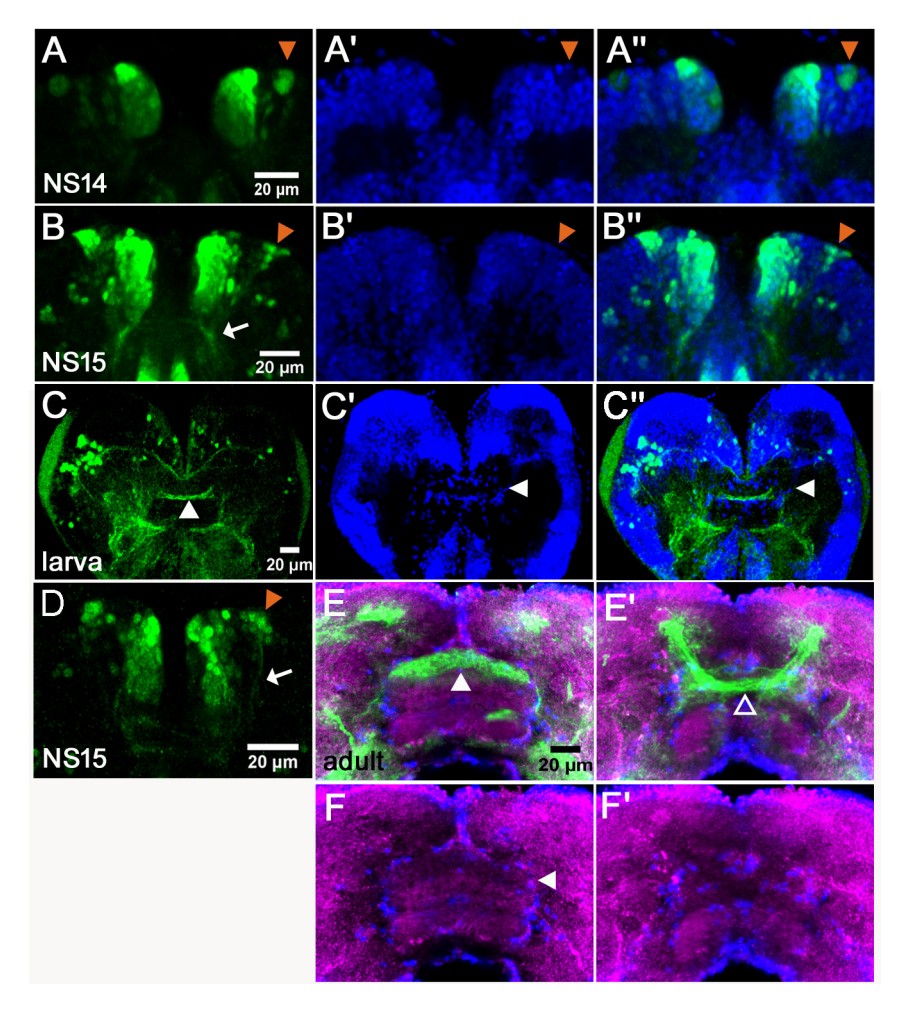

**Figure 5.** *Tc-foxQ2* positive cells project through the central brain primordium and contribute to the central complex. Immunohistochemistry visualizes EGFP (green) derived from the *foxQ2-5'-line*. Synapsin visualizes adult brain morphology (magenta in E,F) while nuclei are visualized by DAPI (blue in A'–C''). (**A–A''**) At NS14, the *anterior-lateral-foxQ2-cluster* consists of one NPC and a small number of progeny (orange arrowheads). (**B–B''**, **D**) At NS15, more cells are marked by EGFP (orange arrowheads) and their projections join a *Tc-foxQ2* positive axon bundle (white arrows in B, (**D**). (**C–C''**) EGFP marked projections contribute to the central body in L5 larval brain, which is visualized by its surrounding glia cells (white arrowhead in C'). (**E–F**) EGFP marked projections contribute to the upper unit of the central body in the adult brain (white arrowhead in E) visualized by synapsin and surrounding glia (white arrowhead in F). (**E'–F'**) Another fascicle projects across the midline directly posterior of the central body (open arrowhead).

DOI: https://doi.org/10.7554/eLife.49065.012

The following figure supplements are available for figure 5:

**Figure supplement 1.** *foxQ2-5'-line* marks cells of the stomodeum and the lateral parts of the labrum.
DOI: https://doi.org/10.7554/eLife.49065.013

**Figure supplement 2.** The *foxQ2-5'-line* marks cells lateral of the stomodeum.
DOI: https://doi.org/10.7554/eLife.49065.014

Both the *anterior-median-foxQ2-cluster* (white arrowheads in *Figure 6E–J*) and the *anterior-lateral-foxQ2-lineage* (orange arrowheads) could be followed throughout development confirming our results in fixed specimen. The full datasets and metadata are available at Zenodo (see Materials and methods).

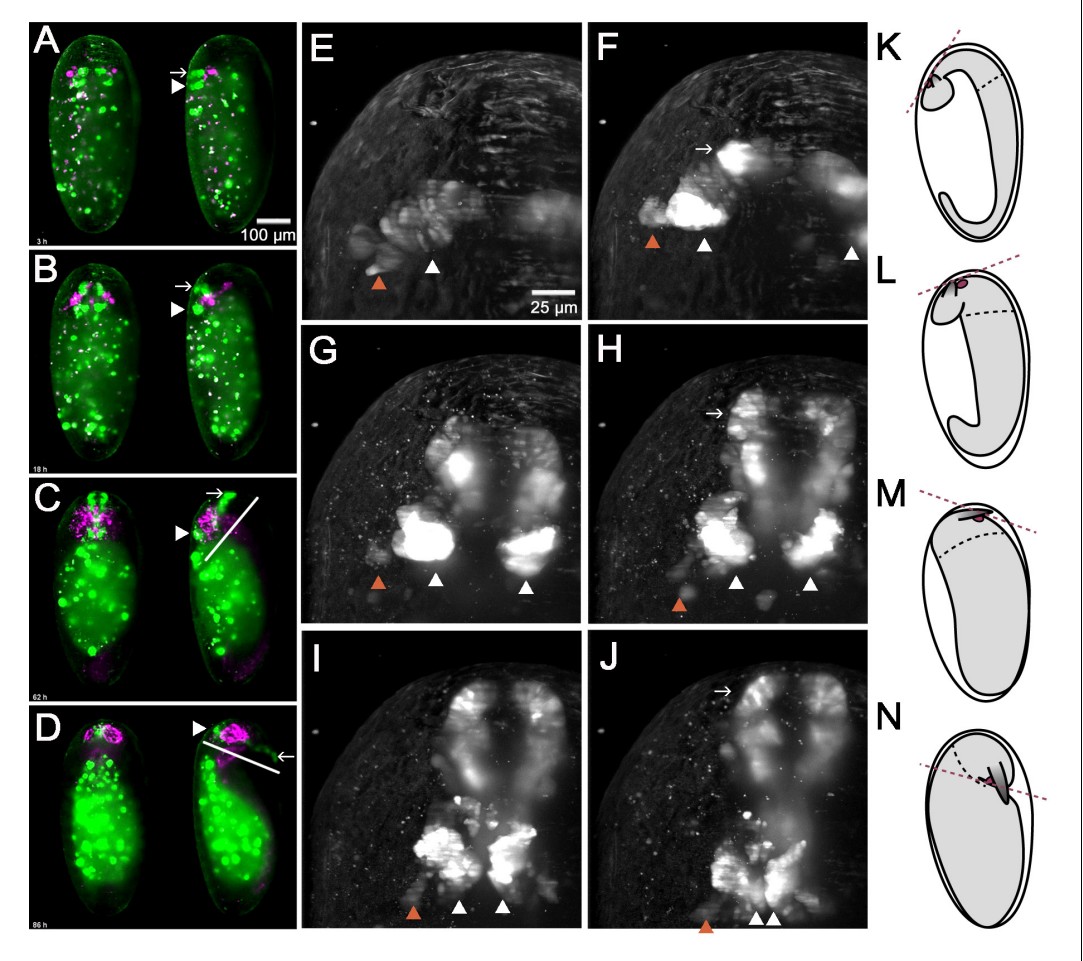

**Figure 6.** In vivo imaging reveals morphogenetic movements during brain development. A cross of the *foxQ2-5'-line* (green) with the *AGOC #6* glia reporter line (magenta) was imaged. (A–D) Shown are dorsal views (left specimen) and lateral views with dorsal to the left (right specimen) (A) EGFP signals in brain and stomodeum start out at the dorsal side (white arrowhead and white arrow, respectively). (B–C) Later, the stomodeum becomes elongated and bends away towards the ventral side. (C–D) An overall ventral bending of the head and brain follows, where the relative positions within the head remain similar (compare white lines in C and D). (E–J) Shown is one specimen from the dorsal side with anterior to the top. At the same time, the bilateral *Tc-foxQ2* positive cell clusters approach each other towards the midline (white arrowheads). Both the *anterior-median-foxQ2-cluster* (white arrowheads in E-J) and the *anterior-lateral-foxQ2-cluster* (orange arrowheads in E-J) can be distinguished throughout development. Note that a small group of marked cells detach from the cluster and fuse at the midline. However, these cells are not Tc-FoxQ2 positive and are, hence, not further considered. (K–N) The overall morphogenetic movements of embryos aredepicted schematically with the embryo highlighted in gray. Marked are the stomodeum (red circle) and the antennae. The black hatched line marks the boundary between thorax and head segments while the red hatched line indicates the plane to which the mouth opens. The embryos are redrawn from *Strobl and Stelzer (2014)* and do not exactly match the stages shown in A-D.

DOI: https://doi.org/10.7554/eLife.49065.015

## Arrest of central brain formation in *Tc-foxQ2* RNAi embryos

Development of the protocerebral brain commissures starts with the midline crossing of the primary commissure pioneer neurons. By subsequent joining of additional neurites from the DM1-4 and other lineages one compact commissure develops. This primary brain commissure later splits and expands into the mature brain commissure, which consists of many midline crossing fascicles (*Boyan et al., 1993*; *Therianos et al., 1995*; *Williams and Boyan, 2008*). Subsequently, formation of the CB by de- and re-fasciculation of axon tracts from a subset of these commissures occurs in grasshopper and *Drosophila* (*Boyan et al., 2017*). Given the contribution of *Tc-foxQ2*[+] cells to the primary commissure and other midline spanning structures, we asked whether it was required for splitting of the primary brain commissure. In order to test this, we knocked down *Tc-foxQ2* function

by RNAi and stained the knock-down embryos with acetylated tubulin (acTub). We found that the primary brain commissure formed but was slightly irregular at NS13 (compare white arrowheads in *Figure 7A and B*). This was in line with our finding that *Tc-foxQ2* positive neurons do not pioneer the commissure but project into it shortly after its formation. Some anterior aberrations stemmed from the previously described loss of the labrum (white arrows in *Figure 7A,B*) (*Kitzmann et al., 2017*). In wildtype NS15 embryos, the brain primordium had increased in size by additional fascicles, the primary commissure had split and first chiasmata had formed (white arrowhead in *Figure 7C*). Strikingly, this process was not observed in *Tc-foxQ2* RNAi animals with strong phenotype. Here, the primary commissure remained compact without signs of splitting (white arrowhead in *Figure 7D*). As a consequence, the brain neuropil remained extremely narrow (compare the space between the cell bodies in *Figure 7C' and D'*, open arrowheads). Lateral to the central body, the

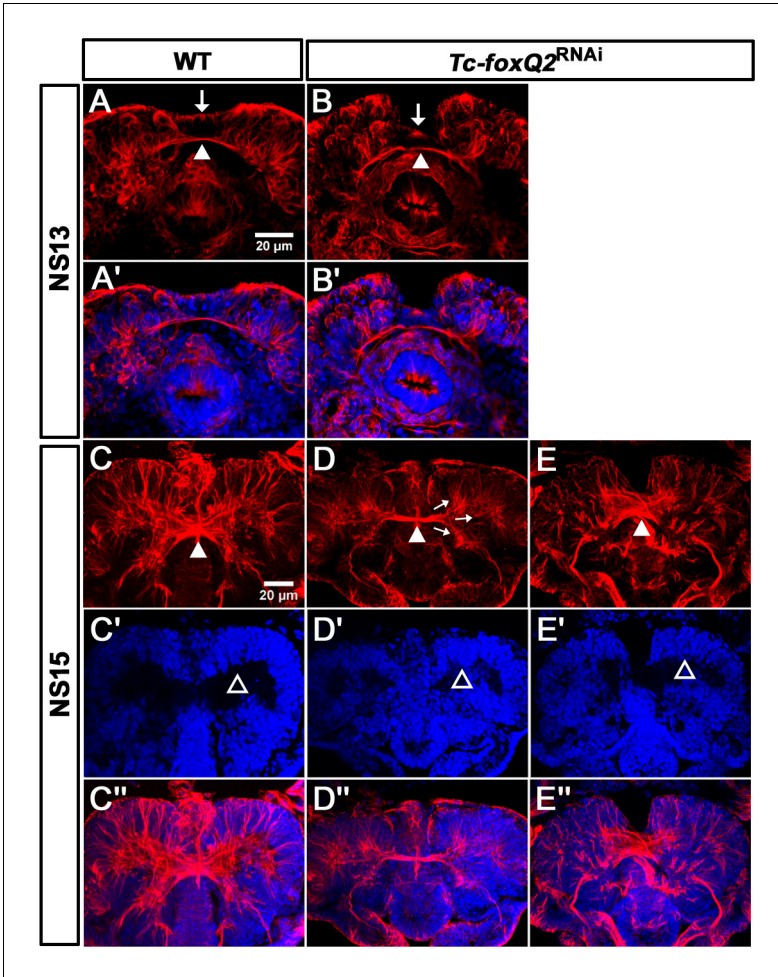

**Figure 7.** Loss of *Tc-foxQ2* function leads to arrest of development of brain midline structures in the embryo. Axonal projections are marked by acTub (red) and cell bodies are visualized by DAPI (blue). (**A, A'**) In WT, the primary brain commissure forms at NS13 (white arrowhead in A). (**B, B'**) In RNAi embryos, the primary brain commissure is slightly irregular (white arrowhead in B). The anterior epidermal aberrations reflect the loss of the labrum (compare white arrows in A and B; *Kitzmann et al., 2017*). (**C–C''**) In WT NS15 embryo, the central brain primordium increases in size and contains more fascicles, some of which form chiasmata at the midline (white arrowhead in C). (**D–D''**) In strong phenotypes, the primary commissure remains detectable along with three main branches (arrows in D). However, the structures do not expand and commissure splitting does not occur. At the same time, the brain neuropil volume is strongly reduced (compare black space between the cell bodies in C' and D', open arrowheads (arrows in D). (**E–E''**). Weak phenotypes show some degree of splitting of the brain commissure but axonal projections are disarranged (white arrowhead in E).
DOI: https://doi.org/10.7554/eLife.49065.016

basic architecture of fascicles emanating from the commissure was mostly intact (e.g. the formation of three main branches emanating from the central brain (arrows in *Figure 7D*)) although it appeared to be built by fewer neurons. Indeed, the number of neurons marked in two imaging lines was reduced (see below). As a consequence, the lateral neuropil area was reduced as well (open arrowheads in *Figure 7D', E'*). In weak phenotypes, commissure splitting had occurred to some degree but the arrangement of fascicles was clearly abnormal (white arrowhead in *Figure 7E*). In summary, *Tc-foxQ2* function is essential for splitting of the brain commissure and for the expansion of the protocerebral neuropil, which together constitute the central brain anlagen. As a consequence, CB development was abolished. Hence, crucial steps of protocerebrum specific features depend on *Tc-foxQ2* function.

## Analyses in novel brain imaging lines reveal a role for Tc-foxQ2 for different lineages

We wondered, in how far different neural lineages would be affected by *Tc-foxQ2* knock-down. To that end, we established two novel transgenic imaging lines that mark subsets of neurons. E035004 is an enhancer trap line generated in the GEKU screen (*Trauner et al., 2009*). We found that the insertion was intragenic in the *Tribolium Teneurin-a* locus (*Ten-a*; TC032747; *Drosophila* synonym: *Tenascin accessory*; CG42338). The EGFP signal overlapped with the cross-reacting *Drosophila* anti-Ten-a antibody staining (*Fascetti and Baumgartner, 2002*). In *Tribolium* embryos, the first midline crossing fascicle was marked while later several fascicles interconnecting the two brain lobes were Ten-a positive (*Figure 8—figure supplement 1*), which is similar to *Drosophila* (*Fascetti and Baumgartner, 2002*). We call this line *Ten-a-green*. In wildtype NS15 embryos three groups of cells were marked. The anterior group contained approximately 39 cells (n = 4; white circle in *Figure 8A*; *Supplementary file 1*-table 7), the posterior-lateral group around 32 cells (n = 4; open arrowhead in *Figure 8A*; *Supplementary file 1*-table 7) and the posterior-median group comprised about 27 cells (n = 4; dashed circle in *Figure 8A*; *Supplementary file 1*-table 7). Importantly for this work, the line marked the central brain primordium. Two *Ten-a* positive fascicles projected across the midline (white arrow in *Figure 8A* marks one of them). These fascicles represented a subset of the acTub positive commissures (compare to *Figure 8A'*). Among other patterns circumesophageal projections were found.

In *Tc-foxQ2* RNAi embryos at NS15, the *Ten-a* positive commissure was highly reduced while the circumesophageal projection was still found. The cell clusters were still discernable but the number of the cells was reduced by half (*Figure 8B*; *Supplementary file 1*-table 7).

Next we generated an enhancer construct by fusing the upstream regulatory region of *Tc-rx* to *dsRedexpress* (*rx-5'-up line*; see Supplementary Materials and methods for details). This line marked an anterior median group of cells, which projected into the central brain (white circle and arrowhead in *Figure 8C*). In addition, a number of peripheral cells without projections into the central brain was marked (open arrowhead in *Figure 8C*). A subset of the marked cells was Tc-Rx positive but a significant number was not (see Supplementary Materials and methods and *Figure 8—figure supplement 1*). Knock-down of *Tc-foxQ2* led to a strong decrease of median cell number at NS15 to about 25% of wildtype (n = 6; *Supplementary file 1*-table 8) and to the complete loss of the marked brain commissures (white circle and arrowhead in *Figure 8D*). The number of peripheral marked cells was reduced as well. In summary, *Tc-foxQ2* knock-down in our imaging lines confirmed the phenotype found in our acTub staining and showed that different neural lineages were affected.

## *Tc-foxQ2* function is required for survival of neural cells

Finally, we asked in how far EGFP expression of the *foxQ2-5'-line* would be affected by knocking down *Tc-foxQ2*. Indeed, at NS13 we found strongly reduced number of cells (less than half; n = 4; *Supplementary file 1*-table 9) in the *anterior-median-foxQ2-cluster* (white arrowheads in *Figure 9A, B*) while the *anterior-lateral-foxQ2-lineage* was either absent or fused to the other cluster. The cells close to the stomodeum were lost completely (not shown). The stomodeal expression, in contrast, appeared unaffected (stars in *Figure 9A,B*). At NS15 the number of neural cells remained less than half of wildtype (n = 4; *Supplementary file 1*-table 9) but the remaining cells always formed connections across the midline. However, these fascicles were thinner and followed an abnormal rounded path instead of the straight line found in wildtype (compare white arrows in *Figure 9E and F*). These

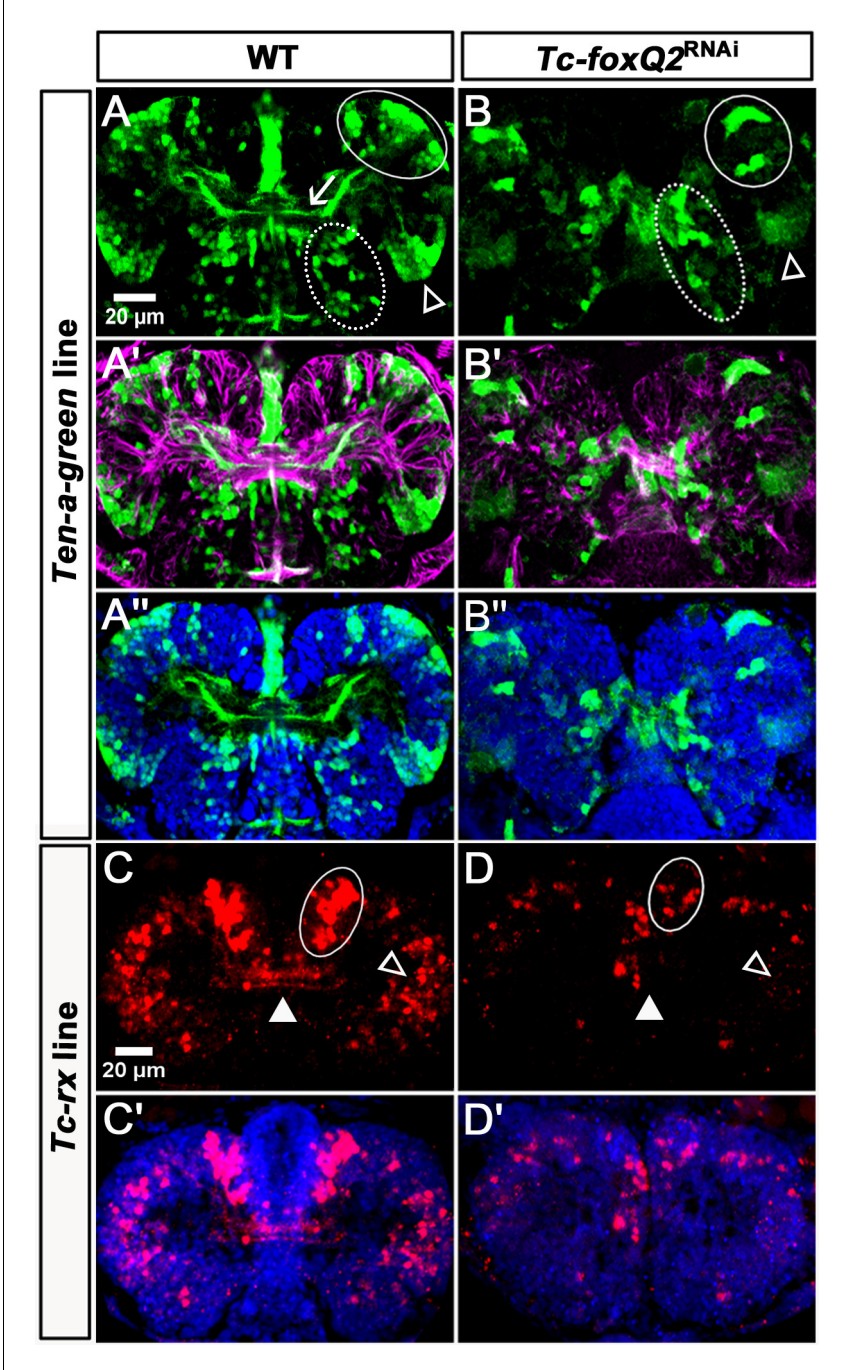

**Figure 8.** Loss of *Tc-foxQ2* function in novel imaging lines confirms the midline phenotype. (**A–A''**) In WT *Ten-a-green* embryos, three groups of cells are marked by EGFP: An anterior group (white circle), a posterior-lateral group (open arrowhead) and a posterior-median group (dashed circle). The central brain primordium is marked with *Ten-a* positive fascicles projecting across the midline (white arrow in A). (**B–B''**) In *Tc-foxQ2* RNAi, the *Ten-a* positive projections and the number of the marked cells is reduced (n = 4). (**C–C'**) In WT *Tc-rx-5'-up* line, the anterior median group of cells marked by DsRed project into the central brain (white circle and white arrowhead). (**D–D'**) In *Tc-foxQ2* RNAi, the cell number in the anterior median group is strongly reduced (n = 6; white circle) and the marked brain commissures are absent (white arrowhead). The peripheral cells are reduced in number as well (n = 6; compare open arrowheads in C,D).

DOI: https://doi.org/10.7554/eLife.49065.017

The following figure supplement is available for figure 8:

*Figure 8 continued on next page*

*Figure 8 continued*

**Figure supplement 1.** Characterization of imaging lines.
DOI: https://doi.org/10.7554/eLife.49065.018

data indicated that *Tc-foxQ2* was required for either survival or specification of neural cells and that upon RNAi these cells were lost or changed their identity. Indeed, increased cell death had been observed after *Tc-foxQ2* RNAi at NS13 (*Kitzmann et al., 2017*). The reduced number of *Tc-foxQ2*⁺ cells was likely the reason for the thinner commissure while its altered path might be a secondary effect due to general misspecification in the central brain after *Tc-foxQ2* RNAi (see below). The reduction of cell number could be due to apoptosis after misspecification of cells due to lack of *Tc-foxQ2*. Alternatively, *Tc-foxQ2* could be involved in an autoregulatory loop, which would lead to EGFP reduction in an RNAi background.

## Discussion

### Does *Tc-foxQ2* mark neuroblasts of both type I and II?

Our data is in line with the hypothesis that type I and type II neuroblasts are present in *Tribolium* and that both types of lineages express *Tc-foxQ2*. Type II neuroblasts were discovered and described recently in *Drosophila* (*Boone and Doe, 2008*; *Bowman et al., 2008*; *Izergina et al., 2009*) and subsequently, lineages with similar properties were described in the grasshopper *Schistocerca gregaria*, which represents the basal hemimetabolan clade of insects (*Boyan et al., 2010*).

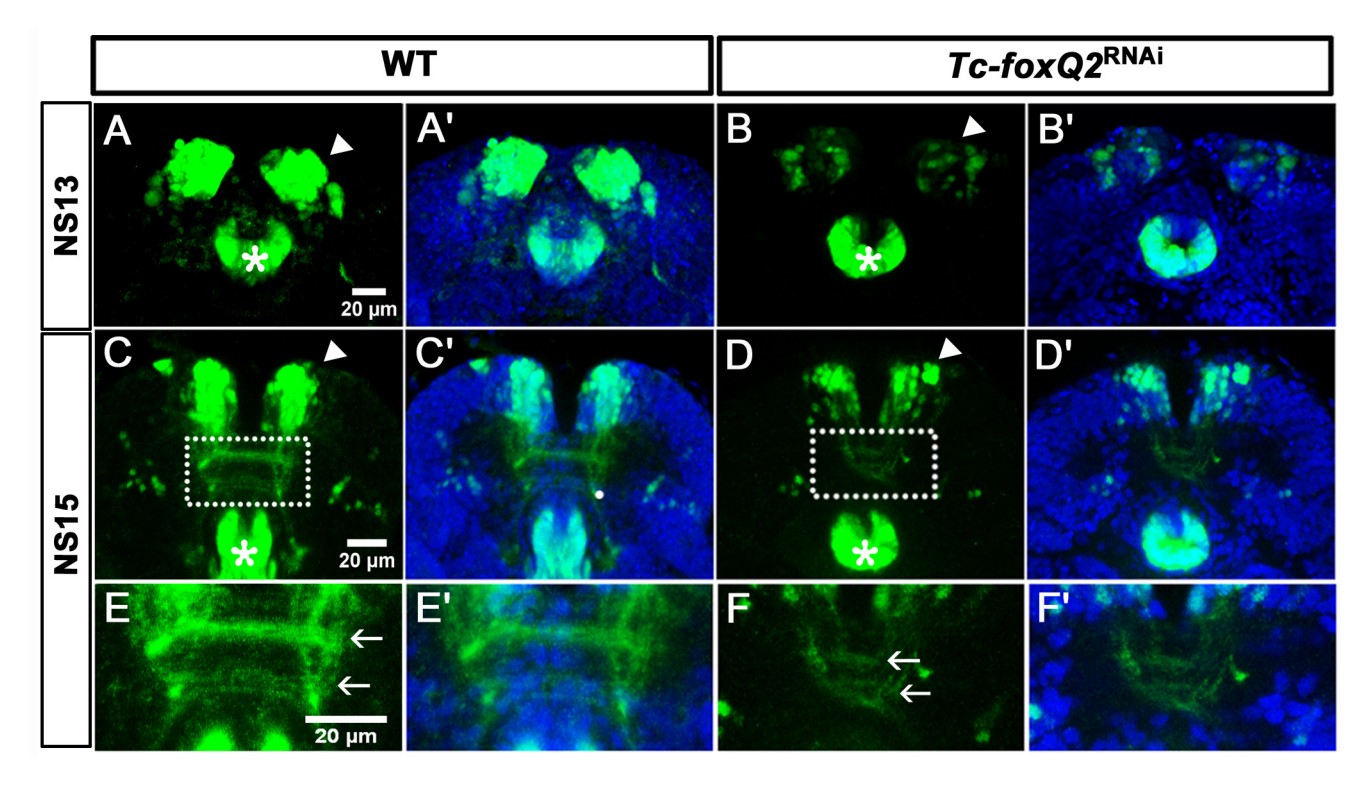

**Figure 9.** RNAi in the *foxQ2-5'-line* indicates self-regulation. (**A–B'**) At NS13, *Tc-foxQ2* RNAi shows the strongly reduced number of marked cells in the *anterior-median-foxQ2-cluster* (n = 4; white arrowheads) while the signal in the stomodeum appears to be unaffected (stars). (**C–F'**) In *Tc-foxQ2* RNAi, the number of marked cells decreased significantly (n = 4; white arrowhead). The fascicles are reduced and follow an abnormal rounded path instead of the straight line in WT (compare white arrows in E and F). Legends figure supplements.
DOI: https://doi.org/10.7554/eLife.49065.019

Therefore, we assume that both types of neuroblasts exist in *Tribolium* as well although this remains to be shown. Type I NBs divide asymmetrically to form ganglion mother cells, which divide once more to form two postmitotic neural cells (*Technau et al., 2006*). All neuroblasts of the ventral nerve cord and most neuroblasts of the brain belong to the type I. Type II neuroblasts, in contrast, give rise to intermediate neural progenitors (INPs), which themselves divide in a stem cell-like fashion to form ganglion mother cells (GMCs) (*Boone and Doe, 2008*; *Bowman et al., 2008*; *Boyan et al., 2010*). This division mode leads to an increased number of cells stemming from one neuroblast. Interestingly, most of the columnar neurons of the central complex derive from type II neural lineages (*Boyan et al., 2010*; *Pereanu et al., 2011*; *Walsh and Doe, 2017*). Unfortunately, molecular markers for reliably distinguishing type I from type II neuroblasts remain to be established in insects outside *Drosophila*. Nevertheless, we suggest that *Tc-foxQ2* marks both types of neuroblasts. The *anterior-lateral-foxQ2-lineage* might be a type I lineage. First, it is located in a lateral region in the neuroectoderm while type II neuroblasts were found in the anterior median brain in other insects (*Boyan and Williams, 2011*; *Walsh and Doe, 2017*). Second, within this cluster we see only one *Tc-ase* marked cell with neuroblast typical morphology throughout several stages of development and the cluster eventually comprises a moderate number of neurons (15-20), which is in the range typical of type I NBs. Finally, this group of cells is surrounded by one glial sheet, which is indicative for neural lineages (*Younossi-Hartenstein et al., 1996*). Using both the location and *Tc-ase* staining pattern as criteria, we assume that this type I lineage might derive from the *P-fox-l* neuroblast (number 3 in *Figure 2I'*).

The *anterior-median-foxQ2-cluster*, by contrast, is likely built by one or more type II neuroblasts. First, the number of *Tc-ase* positive cells in that region decreased over time (*P-fox-am* and *P-fox-amp* in *Figure 2*). This would be rather unusual for neuroblasts but would be expected for INPs and GMCs of a type II lineage, which express *asense* as well (*Álvarez and Díaz-Benjumea, 2018*; *Walsh and Doe, 2017*). Further, the number of *Tc-foxQ2* positive neurons within the single glia niche was much larger (>200 cells), which would be in line with type II mode of division. Moreover, the projection patterns into the CX are reminiscent of the one found in embryonic type II neuroblast lineages in *Drosophila* (*Álvarez and Díaz-Benjumea, 2018*). However, we were not able to unequivocally show a contribution of *Tc-foxQ2* positive cells to the WXYZ tracts, which were described for DM1-4 type II lineages in *Drosophila* and *Schistocerca* (*Boyan and Reichert, 2011*). Hence, this hypothesis needs to be further tested once reliable markers for type II neuroblasts are developed. It would be intriguing to find one regulatory gene contributing to the specification of different types of neuroblasts, which both contribute to the central complex.

The ventrally located group of Tc-foxQ2 positive NPCs probably contributed to a cell cluster close to the stomodeum, which projected into the circumesophageal commissure but not to the central brain (*Figure 5—figure supplement 2*). Due to our focus on central brain development, we did not study this group in detail.

## How does *Tc-foxQ2* function in CX development?

*Tc-foxQ2* positive cells are contributing to the CX and *Tc-foxQ2* is essential for its development. The contribution could affect several stages of CX development. First, the effect could be a consequence of the failure of commissural splitting. In normal development, the primary brain commissure splits into several fascicles, which is the prerequisite for subsequent central body formation. Some neurites undergo fascicle switching, that is they leave their commissure (de-fasciculation) and bridge to another commissure, which they join (re-fasciculation). As consequence, these neurites form X-shaped crossings prefiguring the columns of the central body (*Boyan and Williams, 2011*; *Boyan et al., 2008*; *Boyan et al., 2017*). In the absence of commissural splitting, this process and central body formation cannot take place. A putative second contribution of *Tc-foxQ2* could be to specify the points of fascicle switching. In the late embryonic brain, *Tc-foxQ2* marks commissures that are located in the region where this process will occur. In this model, Tc-foxQ2 positive neurites could either be the ones that de-fasciculate or they could be required to provide the signal for other neurites to do so. Indeed, we find *Tc-foxQ2* positive fascicles closely associated with the larval central body. Due to the arrest of development after the first phenotype, the subsequent processes cannot be observed and remain to be tested. Scrutinizing the development of individual *Tc-foxQ2* positive cells would be helpful. Actually, we have designed our enhancer trap line to expresses Cre

along with EGFP. Therefore, once a transgenic *Tribolium* line with a functional brainbow construct is developed, single neurons within the *Tc-foxQ2* positive clusters can be traced (*Livet et al., 2007*).

## Making the protocerebrum different from segmental ganglia

Our results show that *Tc-foxQ2* is one of the regulatory genes required for the development of structures that distinguish the protocerebrum from segmental ganglia. Specifically, it is involved in the unique development of the brain commissures and the midline spanning neuropils of the central complex. We found several protocerebrum-specific functions. First, *Tc-foxQ2* is expressed in neural progenitors of the protocerebrum but not in the more posterior ganglia. Hence, *Tc-foxQ2* has indeed the potential of contributing to developmental programs that are specific to this brain part. Therefore, this gene adds to the list of co-expressed genes assumed to specify NB identity in the *Drosophila* brain (*Urbach and Technau, 2003a*). A second protocerebrum-specific role is the contribution of *Tc-foxQ2* positive neurons to the CX, which is a strictly protocerebral neuropil. Third, *Tc-foxQ2* positive neurons mark subsets of axons within the early brain commissure ending up in different commissural fascicles after the split of the primary commissure.

## Evolutionary relationship of apical organ and the central complex

*foxQ2* together with *six3* and other genes are part of the anterior gene regulatory network (aGRN) in animals and they contribute to the development of anterior-most neural structures. Specifically, they are involved in patterning the apical organ/apical tuft including serotonergic and neurosecretory cells in marine larvae, cnidarians, annelids and sea urchins (*Leclère et al., 2016*; *Marlow et al., 2014*; *Yaguchi et al., 2008*). However, apical organs are usually lost during metamorphosis and clear morphological homologs were not found in neither insects nor vertebrates (*Nielsen, 2005*). However, correlation of the aGRN with neuroendocrine cells was found in animals of all phyla including arthropods (*Hunnekuhl and Akam, 2014*; *Oliver et al., 1995*; *Posnien et al., 2011b*; *Steinmetz et al., 2010*). It has been proposed, that this aGRN derives from an ancestral apical brain (with neuroendocrine and non-visual light detection functions), which fused with a more posterior blastoporal brain (with sensory-contractile function) to form the 'chimeric' anterior brain of bilaterians (*Tosches and Arendt, 2013*). Unfortunately, the lack of a clear apical organ homolog in insects has hampered the comparison of the respective brain parts based on morphology. Hence, molecular similarity has to be interrogated instead for such analyses. Indeed, it was found before that *six3* and *rx* are required for the formation of central brain structures with respect to the expected neuroendocrine cells but also for other protocerebral structures like mushroom bodies and central complex (*Davis et al., 2003*; *de Velasco et al., 2007*; *Eggert et al., 1998*; *Kraft et al., 2016*; *Kunz et al., 2012*; *Posnien et al., 2011b*; *Steinmetz et al., 2010*). The role of *Tc-six3* and *Tc-foxQ2* in *Tribolium* brain development had been noted but a detailed analysis of the respective brain phenotypes had not been performed and importantly, the projection patterns of these cells had not been studied in any insect (*Kitzmann et al., 2017*; *Posnien et al., 2011b*).

In this work we used *Tc-foxQ2* as a very specific marker for a subset of cells deriving from the aGRN region (*Kitzmann et al., 2017*). Based on its limited and highly conserved expression throughout bilaterian and its functional interconnection with *six3*, *foxQ2* is likely to represent a specific marker for the hypothetical apical brain. As expected, the cell bodies of *Tc-foxQ2* positive cells are located in the pars intercerebralis, which is one of the neuroendocrine parts of the brain, being marked by *six3* expression (*Steinmetz et al., 2010*). Hence, this is another piece of evidence supporting the ancestral neuroendocrine function of the apical region. Unexpectedly, our data revealed for the first time a contribution of these cells to the brain commissure and to the CX. These represent important non-neuroendocrine structures required for motor control, which would be indicative for an origin from the suggested blastoporal brain with its sensory-contractile origin (*Tosches and Arendt, 2013*). This was unexpected given that cells derived from the apical brain were thought to be mainly involved in neuroendocrine control and non-visual photoperception.

There are two evolutionary scenarios for the involvement of an apical neural cell to motor control. The chimeric brain hypothesis states that apical and the anterior part of the blastoporal brain fused to form the anterior brain of bilaterians. In the light of this hypothesis, these cells could represent the predicted chimeric cells combining features from both the neuroendocrine apical brain and the motor control function of the blastoporal brain (*Tosches and Arendt, 2013*).

Alternatively, this result could indicate that *foxQ2* positive cells evolved novel functions after the apical organ became reduced in the lineage leading to arthropods. In this model, the reduction of the apical organ made the *foxQ2* positive cells free to evolve and they integrated into the evolving CX. Actually, a canonical CX is found only in crustaceans while homology of brain midline structures of other arthropods with the CX remain equivocal (*Hanström, 1928*; *Holmgren, 1916*; *Loesel et al., 2002*; *Loesel et al., 2011*; *Strausfeld, 2012*). A contribution of the apical organ to brain development has been suggested in the annelid *Platynereis dumerilii*, where the apical organ may provide an initial scaffold for the developing anterior brain (*Marlow et al., 2014*). In the millipede *Strigamia maritima*, *foxQ2* positive cells located at the brain midline build an early axonal scaffold forming a midline brain structure from which longitudinal tracts project bilaterally into the trunk (*Hunnekuhl and Akam, 2014*). Based on these findings, we suggest a model, where in a bilaterian arthropod ancestor the apical organ provided a scaffold for subsequent neural development of the anterior brain. In basal arthropods like myriapods, the initially simple midline structure with contribution from the apical organ expanded to form a prominent midline structure, which was still of low complexity. The respective neurons provided a more prominent scaffold for subsequent neural development. In crustaceans, this simple architecture was further developed to build a more elaborate CX, which in stomatopods and insects reached its most complex realization (*Thoen et al., 2017*). A comparison of the molecular signatures of *foxQ2* positive cells in animals might be helpful to trace back their evolutionary trajectories.

## Outlook

*foxQ2* is one of several genes with an almost exclusive anterior expression in animals. We show its crucial contribution to protocerebrum-specific development. Based on this, it seems imperative that other highly conserved and protocerebrum-specific genes be considered when investigating brain development (e.g. *rx*, *six3*, *chx*, *fez1*, etc.; *Kitzmann et al., 2017*; *Posnien et al., 2011b*). An approach focusing on genes known from the ventral nerve cord might fall short of important insights. Given the rather exclusive contribution of foxQ2 positive neurons, it will be important to determine their individual projection patterns in both *Drosophila* and *Tribolium* in order to map them with identified neurons from existing resources and to determine the degree of conservation. Importantly, our concept of genetic neural lineages turned out to work quite well and opens the way for comparative studies (*Koniszewski et al., 2016*). As a basis it will be necessary to systematically compare cell types and markers between *Drosophila* and other insects in order to ascertain that orthologous genes mark homologous cell types. Finally, the evolution of *foxQ2* positive neurons provide an interesting study case for neural evolution: From a role in patterning a rather simple structure, the apical organ, they expanded their role in arthropods contributing to one of the most intricate insect brain structures. On the other hand, *foxQ2* was lost in amphibians and mammalians, which is astonishing regarding the usually high degree of conservation of anterior regulators.

## Materials and methods

### Key resources table

| Reagent type (species) or resource | Designation | Source or reference | Identifiers | Additional information |
|---|---|---|---|---|
| Gene (*Tribolium castaneum*) | *foxQ2* | iBeetle-Base http://ibeetle-base.uni-goettingen.de/ | TC004761 | *Drosophila* Ortholog: *fd102*, CG11152 |
| Strain, strain background (*Tribolium castaneum*) | *San Bernardino* | | *SB* | NCBI:txid7070 |
| Strain, strain background (*Tribolium castaneum*) | *vermillion*[white] | | *vw* | NCBI:txid7070 for transgenesis, mutant eye color (white) is rescued to black by 3XP3-vw |

*Continued on next page*

*Continued*

| Reagent type (species) or resource | Designation | Source or reference | Identifiers | Additional information |
|---|---|---|---|---|
| Genetic reagent (*Tribolium castaneum*) | *foxQ2-5'-line* | this publication | | marks Tc-foxQ2 positive cells with EGFP; maintained by Bucher-Lab |
| Genetic reagent (*Tribolium castaneum*) | *Ten-a-green-line* | this publication | | marks Ten-a positive cells with EGFP; maintained by Bucher-Lab |
| Genetic reagent (*Tribolium castaneum*) | *rx-5'-up-line* | this publication | | marks a subset of Tc-rx positive cells with dsRedexpress; maintained by Bucher-Lab |
| Recombinant DNA reagent | [3xP3:Tc'v-SV40 -Cre-2A-EGFP: bhsp68-eb] | this publication Addgene plasmid #124068 | | repair template for NHEJ mediated enhancer traps in *Tribolium castaneum* EGFP and Cre under the control of the *Tribolium* core heat-shock promoter, which is not heat-shock responsive but takes up enhancer traps. |
| Recombinant DNA reagent | bhsp68-Cas9 | *Gilles et al., 2015* Addgene plasmid #65959 | | Cas9 gene for co-injection |
| Antibody | anti-GFP (chicken polyclonal) | Abcam | RRID:AB_300798 | 1:1000 |
| Antibody | anti-acetylated Tubulin (mouse monogclonal) | Sigma Aldrich | RRID:AB_609894 | 1:50 |
| Antibody | anti-SYNORF1 (synonym: anti synapsin) (mouse monoclonal) | DHSB Hybridoma Bank (University of Iowa) | RRID:AB_2313867 3C11 (DSHB ID) | 1:40 |
| Antibody | anti-RFP (rabbit polyclonal) | Abcam, ab62341 | RRID: AB_945213 | 1:1000 |
| Antibody | anti-Tenascin (Teneurin)-a (rabbit polyclonal) | *Fascetti and Baumgartner, 2002* | | 1:1000 |
| Antibody | Secondary antibodies coupled with Alexa Fluor 488 or Alexa Fluor 555 (goat anti mouse or goat anti chicken, polyclonal) | Thermo Fisher Scientific Cat # A32723 Cat # A32727 Cat # A-11039 Cat # A-21437 | RRID:AB_2633275 RRID:AB_2633276 RRID:AB_142924 RRID:AB_2535858 | 1:1000. |
| Commercial assay or kit | MEGAscript T7 Transcription Kit | Thermo Fisher Scientific | | production of dsRNA |
| Sequence-based reagent | *Tc-foxQ2*[RNAi_a]-L | *Kitzmann et al., 2017* | | Primer for dsRNA template production: GAATTGTAATACGACTCAC TATAGGCTTACTTCAGGACCCGG |
| Sequence-based reagent | *Tc-foxQ2*[RNAi_a]-R | *Kitzmann et al., 2017* | | Primer for dsRNA template production: GAATTGTAATACGACTCACT ATAGGTCGCTTG TAACAATGCTTGA |

## Animals

*Tribolium castaneum* NCBI:txid7070 animals were reared under standard conditions at 32°C. The *San Bernadino* (*SB*) wild-type strain was used for fluorescent in situ hybridization, antibody staining and RNAi experiments. The *Tc-vermillion*[white] (*Tc-vw*) strain (*Lorenzen et al., 2002*) was used for transgenesis.

## Generation of a tc-foxq2 polyclonal antibody

The C-terminus (amino acids 202–286) was amplified from cDNA by PCR using primer pairs with *BsaI* restriction site forward and reverse (For primer sequences see *Supplementary file 1*-table 10) and cloned into pET SUMO vector generating a fusion protein with a His-SUMO tag using golden gate cloning (modified from Thermo Fischer). The protein was expressed in BL21-DE3 Rosetta cells at 37° C. Cells were fractionated (50 mM TRIS-HCl pH 7.8, 500 mM NaCl, 10 mM Imidazole) using Fluidizer (mechanical lysis by high pressure-80 psi). The protein was purified via $Ni^{2+}$ chelate affinity chromatography using a gradient with 200 mM imidazole in lysis buffer. Dialysis (50 mM Tris-HCl pH 7.8, 500 mM NaCl) and SUMO protease digestion for cleavage of the His-SUMO tag were performed simultaneously overnight. The His-SUMO tag was removed from the Tc-FoxQ2 via Re-$Ni^{2+}$ chelate affinity chromatography. Gel-filtration chromatography (Superdex G-30 Healthcare) was performed to remove the remaining contaminations and finally the purified Tc-FoxQ2 was stored in phosphate-buffered saline (PBS). All the steps for purification were done at 4°C. Antibodies were produced in guinea pigs by Eurogentec (Liège, Belgium). The final serum was used straight as the Tc-FoxQ2 antibody with the dilution of 1:1000. Before antibody staining, pre-absorption of anti-FoxQ2 was performed for eliminating non-specific binding.

## Generation of imaging lines and stocks

### foxQ2-5'-line

The guide RNAs (gRNAs) were designed with the aid of the flyCRISPR Optimal Target Finder (http://tools.flycrispr.molbio.wisc.edu/targetFinder/; *Gratz et al., 2014*). The TriGenes gRNA oligo design tool was used for generating the sequences of the oligos to order. The annealed oligos were cloned into the gRNA expression vector p(TcU6b-*BsaI*) via the BsaI restriction sites. The detailed annealing and ligation are following the protocol described previously (*Gilles et al., 2015*). [3xP3: Tc'v-SV40-Cre-2A-EGFP:bhsp68-eb] was designed as a repair template for NHEJ-mediate knock-in by CRISPR/Cas9. For linearizing the plasmid, *Dm-ebony* target site (gRNA-eb) was cloned into this construct (Addgene plasmid # 124068). Each fragment of the construct was amplified by PCR from plasmids available in the laboratory's plasmid library (*Supplementary file 1*-table 3) by using primers with overhangs that are complements of two adjacent fragments (*Supplementary file 1*-table 10). The 2A-peptide and the target sequences for cleavage were completely added by primers. Overlap extension PCR was performed to assemble all fragments together. In addition, an *ApaI* and a *XbaI* restriction sites were added in end primers for the following ligation. The entire construct was finally cloned into pJET1.2 vector. The helper plasmid p(bhsp68-Cas9) expressing Cas9 was a gift from Michalis Averof (Addgene plasmid # 65959).

Embryonic injection was performed in *Tc-vermillion*[white] (*Tc-vw*) according to standard procedure (*Berghammer et al., 1999*; *Schinko et al., 2012*). Two gRNAs (gRNA1 and gRNA2) targeting the upstream region of *Tc-foxQ2* together with the repair plasmid, Cas9 expression plasmid and gRNA-eb were injected. The final concentration of Cas9 plasmid and the repair plasmid is 500 ng/μl each, and gRNA is 125 ng/μl each. The injected animals were separated into male and female during pupal stage. Each animal was crossed to three *Tc-vw* wild type beetles of the opposite sex. The G1 offspring were screened for black eyes. The transgenic beetle was outcrossed with *Tc-vw* wild type and kept as a new stock.

### rx-5'-up line

The regulatory region including the endogenous promoter sequence of the gene *Tc-rx* was amplified from genomic DNA by PCR using primer pairs with restriction sites *BamHI* and *NheI* (for primer sequences see *Supplementary file 1*-table 10) and cloned into the Dual Promoter pCRII vector by using the TA Cloning Kit (Invitrogen). The final construct was designed with the following sections from 5′ to 3′: (1) regulatory regions, (2) endogenous promoter, (3) reporter gene. The reporter gene

DsRedExpress was amplified from the other construct by using according primers (*Supplementary file 1*-table 10). The construct was designed and created in the vector pslfa1180fa. Afterwards, the cassette [rx-5'up:DsRedEx-SV40] including the regulatory region, promoter, reporter gene, and SV40 was transferred into the piggyBac[3xP3:Tc'v-SV40]fa transformation vector by using the restriction enzymes *AscI* and *FseI* (New England BioLabs). Further steps and treatments for embryonic transgenesis were performed as described (*Berghammer et al., 1999*; *Schinko et al., 2012*).

## RNAi

Both dsRNA fragment and parental injection was performed as in *Kitzmann et al. (2017)*, where off target controls had been performed. The injected dsRNA concentrations were 1.5 µg/µl and 3.0 µg/µl. 250–300 pupae were injected and >50 offspring embryos were fixed and stained, respectively. In larval and adult experiments, we stained >5 brains. 4–6 brains were analyzed by LSM imaging. We only considered phenotypes that we saw in several independent specimen.

## Immunhistochemistry and FISH

Immunostaining of embryonic, larval and adult brains were performed according to the described protocol (*Büscher et al., 2019*; *Hunnekuhl et al., 2019*). FISH was performed using a horseradish peroxidase (POD) mediated tyramide signal amplification (TSA). Primary antibodies: chicken anti-GFP (1:1000, Abcam; RRID:AB_300798), mouse anti-acetylated Tubulin (1:50, Sigma; RRID:AB_609894), mouse anti-Synapsin (1:40, DHSB Hybridoma Bank; RRID:AB_2313867), rabbit anti-RFP (1:1000, Abcam; RRID: AB_945213), polyclonal rabbit anti-Tenascin (Teneurin)-a (*Fascetti and Baumgartner, 2002*). Secondary antibodies coupled with Alexa Fluor 488 or Alexa Fluor 555 (Thermo Fisher Scientific, RRID:AB_2633275 RRID:AB_2633276 RRID:AB_142924 RRID:AB_2535858) were used at 1:1000.

## In vivo imaging

Long-term live imaging was performed with digitally scanned laser light sheet-based fluorescence microscopy (DSLM, LSFM) as described previously for *Tribolium* (*Strobl et al., 2015*; *Strobl et al., 2017*). In brief, embryos were collected either (i) from a homozygous *foxQ2-5'* culture or (ii) from two hybrid cultures that consisted either of homozygous *foxQ2-5'* females mated with (mO-mC/mO-mC) homozygous AGOC #6 males or of (mO-mC/mO-mC) homozygous AGOC #6 females mated with homozygous *foxQ2-5'* females. After one hour of collection at 25°C, embryos were incubated for 20 hr at 32°C. Sample preparation took approximately one hour at room temperature (23 ± 1°C), so that embryos were at the beginning of germband retraction. Embryos were recorded either (i) only along the dorsal axis or (ii) along the dorsal and lateral axis with an interval of 60 min. All shown embryos survived the imaging procedure, developed to healthy and fertile adults, and when mated either (i) with a homozygous *foxQ2-5'* sibling or (ii) with a (mO-mC/mO-mC) homozygous AGOC #6 sibling, produced progeny that was also fertile. Each setup for imaging this process (both 40X and 10X) was done twice summing up to four independent documentations. Metadata of the three datasets is provided with the Zenodo dataset.

## Image processing and documentation

Immunohistochemistry and FISH were imaged using a ZEISS laser scanning microscope LSM510. Stacks were processed using ImageJ (v.1.47). Images were level-adjusted for brightness and contrast and assembled in Photoshop CS (Adobe). The stacks are available in both original Zeiss LSM format and as avi on figshare (https://figshare.com/account/home#/projects/62939).

# Acknowledgements

We thank Natalia Carolina García Pérez and Rebekka Wallrafen for mapping and initial characterization of the *Ten-a line* and Achim Dickmanns' (GZMB) help with the purification of the protein. Kolja Eckermann for providing the SUMO plasmid and Vera Hunnekuhl for valuable comments on the manuscript. In vivo imaging was performed with the infrastructure provided by Ernst Stelzer. Elke Küster helped identifying and keeping the transgenic stocks. The Ten-a antibody was kindly

provided by Stefan Baumgartner. Themonoclonal antibody anti SYNORF1 was obtained from the Developmental Studies Hybridoma Bank, created by the NICHD of the NIH and maintained at The University of Iowa, Department of Biology, Iowa City, IA 52242. We acknowledge support by the Open Access Publication Funds of the Göttingen University.

## Additional information

### Funding

| Funder | Grant reference number | Author |
|---|---|---|
| Deutsche Forschungsgemeinschaft | BU1443/10 | Gregor Bucher |
| China Scholarship Council | 201406350036 | Bicheng He |

The funders had no role in study design, data collection and interpretation, or the decision to submit the work for publication.

### Author contributions

Bicheng He, Conceptualization, Data curation, Funding acquisition, Investigation, Visualization, Methodology, Writing—original draft, Writing—review and editing; Marita Buescher, Data curation, Supervision, Methodology, Writing—review and editing, Developed the visualization of neuroblasts by intronic ase staining; Max Stephen Farnworth, Investigation, Methodology, Writing—review and editing; Frederic Strobl, Investigation, Visualization, Methodology, Writing—original draft; Ernst HK Stelzer, Resources, Supervision, Methodology; Nikolaus DB Koniszewski, Investigation, Methodology, Developed the transgenic rx-imaging line; Dominik Muehlen, Methodology, Developed the enhancer trap system; Gregor Bucher, Conceptualization, Resources, Data curation, Supervision, Funding acquisition, Visualization, Methodology, Writing—original draft, Project administration, Writing—review and editing

### Author ORCIDs

Max Stephen Farnworth [iD] http://orcid.org/0000-0003-2418-3203
Ernst HK Stelzer [iD] https://orcid.org/0000-0003-1545-0736
Gregor Bucher [iD] https://orcid.org/0000-0002-4615-6401

### Decision letter and Author response

Decision letter https://doi.org/10.7554/eLife.49065.031
Author response https://doi.org/10.7554/eLife.49065.032

## Additional files

### Supplementary files

• Supplementary file 1. Supplementary tables including quantifications and oligo sequences.
DOI: https://doi.org/10.7554/eLife.49065.020
• Transparent reporting form DOI: https://doi.org/10.7554/eLife.49065.021

### Data availability

All LSM stacks can be downloaded from the figshare repository (https://figshare.com/projects/Additional_Data_for_He_et_al_foxQ2_is_required_for_protocerebrum_specific_brain_development_and_marks_cells_of_the_central_complex_in_the_beetle_Tribolium_castaneum_/62939). The construct used for generating the enhancer trap is available from AddGene (#124068). The in vivo imaging data is accessible at Zenodo (http://doi.org/10.5281/zenodo.2645645; http://doi.org/10.5281/zenodo.2645657; http://doi.org/10.5281/zenodo.2645665).

The following datasets were generated:

| Author(s) | Year | Dataset title | Dataset URL | Database and Identifier |
|---|---|---|---|---|
| He B, Buescher M, Farnworth MS, Strobl F, Stelzer E, Koniszewski NDB, Mühlen D, Bucher G | 2019 | In vivo imaging of foxQ2 postitive neurons in the beetle Tribolium castaneum (10X) | http://doi.org/10.5281/zenodo.2645645 | Zenodo, 10.5281/zenodo.2645645 |
| He B, Buescher M, Farnworth MS, Strobl F, Stelzer E, Koniszewski NDB, Mühlen D, Bucher G | 2019 | In vivo imaging of foxQ2 postitive neurons in the beetle Tribolium castaneum (40X) | http://doi.org/10.5281/zenodo.2645657 | Zenodo, 10.5281/zenodo.2645657 |
| He B, Buescher M, Farnworth MS, Strobl F, Stelzer E, Koniszewski NDB, Mühlen D, Bucher G | 2019 | In vivo imaging of foxQ2 postitive neurons in the beetle Tribolium castaneum | http://doi.org/10.5281/zenodo.2645665 | Zenodo, 10.5281/zenodo.2645665 |
| Bicheng He, Marita Buescher, Max Stephen Farnworth, Frederic Strobl, Ernst HK Stelzer, Nikolaus DB Koniszewski, Dominik Muehlen, Gregor Bucher | 2019 | Additional Data for He et al. "foxQ2 is required for protocerebrum specific brain development and marks cells of the central complex in the beetle Tribolium castaneum" | https://figshare.com/projects/Additional_Data_for_He_et_al_foxQ2_is_required_for_protocerebrum_specific_brain_development_and_marks_cells_of_the_central_complex_in_the_beetle_Tribolium_castaneum_/62939 | figshare, 62939 |

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
