## [Decision Letter]

Thank you for submitting your article "An ancestral apical brain region contributes to the central complex under the control of *foxQ2* in the beetle *Tribolium*" for consideration by *eLife*. Your article has been reviewed by three peer reviewers, and the evaluation has been overseen by a Reviewing Editor and Patricia Wittkopp as the Senior Editor. The following individual involved in review of your submission has agreed to reveal their identity: Angelika Stollewerk (Reviewer #1).

The reviewers have discussed the reviews with one another and the Reviewing Editor has drafted this decision to help you prepare a revised submission.

As you will see from the reviews below, the reviewers do not ask for any additional experiments as they are happy with the thoroughness of the approach that you took. However, all three reviewers want you to revise the manuscript to make it more accessible, but also to present more background to allow a wider audience to access a manuscript that might appear otherwise esoteric, in part due to the non-conventional system (*Tribolium* is not yet a full blown model organism).

Therefore, we think that you will need to do a more thorough job in presenting your data, and in particular the comparative context (as precisely outlined in the three reviews). One of the reviewers also believes that you must provide a much better introduction of the central complex and of the whole apical nervous system idea. But new experiments are not required.

Reviewer #1:

The manuscript analyses the expression and role of *Tribolium foxQ2*, a conserved developmental gene, which has been associated with the formation of anterior-most neural structures across the animal kingdom. The authors show that in *Tc-foxQ2* RNAi embryos the protocerebral commissure does not become subdivided and as a consequence the central complex does not form.

Overall the manuscript is well written. The figures are excellent and support the narrative. The Discussion is concise and puts the main findings into context. However, the authors seem to directly translate knowledge of developmental processes based on studies in other insects (e.g. grasshopper, *Drosophila*) to *Tribolium* (e.g. type I and II neuroblasts, fasciculation patterns; details below) without clarifying that confirmation of similar processes in *Tribolium* are lacking.

Furthermore, the authors discuss what makes the protocerebrum different with regard to the segmental ganglia and state: "The third and most striking role of *Tc-foxQ2* in making the protocerebrum different from the segmental ganglia is the requirement for commissure splitting." This statement is not true. In *Drosophila*, the two segmental commissures of the ventral nerve cord are initially established as one commissure, which is then split into an anterior and posterior commissure by the migration of the midline glia cells. The commissures are subsequently further divided into tracts along the dorso-ventral axis (again by the midline glia).

Finally, the analysis of the brain imaging lines would be more meaningful, if the authors correlated the expression with developing brain centers.

Reviewer #2 (General assessment and major comments (Required)):

This paper provides novel information about the formation of the central complex in *Tribolium*, and the role of the transcription factor *FoxQ2* in this process. The paper makes a significant contribution in pushing forward the detailed neuroanatomical analysis of brain development in *Tribolium*, providing novel tools very useful for the community, and new insights into how the arthropod brain is specified and differentiates. The text is generally well written and illustrated. Comments:

1) Results, subsection “Marking of the genetic *Tc-FoxQ2* neural lineage by a CRISPR/Cas9 induced enhancer trap”

How do the medial and lateral domain described here for antibody and EGFP match the three groups of neural progenitors introduced in subsection “*Tc-foxQ2* marks neural progenitor cells with four different molecular identities”? Potential differences in EGFP expression should be mentioned in more detail, referring to the three previously described groups.

2) Subsection “Three *Tc-FoxQ2* positive cell clusters contribute to brain midline structures including the central body”

From the description it becomes clear that the *foxQ2* Gal4 line remains expressed longer than the gene itself (referring to subsection “*Tc-foxQ2* marks neural progenitor cells with four different molecular identities”). The authors should describe this more explicitly. Do the number of lineages that are speculated to form part of the antero-median cluster match the number of progenitors seen earlier (by antibody) in the anteromedial cluster?

3) Subsection “Three *Tc-FoxQ2* positive cell clusters contribute to brain midline structures including the central body”

At the beginning of the section (NS13) the authors propose that the anteromedian clusters contains multiple lineages (second paragraph); at the end, based on glial marker, it sounds like that the cluster corresponds to a single lineage (fifth paragraph). Please clarify.

4) Subsection “Arrest of central brain formation in *Tc-foxQ2* RNAi embryos”

The terms/concept concerning the "primary commissure" and "splitting of the commissure" need to be clarified. What exactly is defined here as the "primary commissure"? Up until which stage do the authors call the crossing fibers "primary commissure"? Is it a single bundle, or more than one?

Concerning the "splitting": It is not clear how the pattern of labeling, with Tubulin antibody or the *foxQ2*-Gal4, relates to the reality of commissure formation. None of the markers will label all fibers (otherwise, the entire neuropil would be homogenously labeled). There are some fibers/fiber tracts that, at a given time point, will label, whereas others will not (or more weakly). Thus, if one initially sees one single labeled commissural tract, and later two such tracts, separated by a non-labeled "cleft": this could mean (1) that a single commissural bundle has grown in number of its fibers (because neuron of the same class/lineage successively join the bundle), and then a central subset of its fibers (the first born ones?) have lost expression of the marker; or (2) that there was one bundle which does not change, and a second bundle (from a separate class/lineage) has grown alongside, separated by the first by a non-neural cleft (glial sheath?). These distinctions are important when characterizing the RNAi phenotype.

It should also be addressed why the "strong" phenotype resulted in such a massive loss of fiber tracts/neuropil. Was the number of cell bodies (e.g., volume of cortex) equally reduced? Would that fit with the fairly restricted expression of *foxQ2* in a few lineages?

5) Subsections “Analyses in novel brain imaging lines reveals a role for *Tc-foxQ2* for different lineages” and “*Tc-foxQ2* function is required for survival of neural cells”

As indicated above, the authors should address the effect of *foxQ2* RNAi on neural lineages in generally more explicitly. If it is true that lineages outside the *foxQ2*-expressing domain are affected, this would imply some inductive effect of the anterior-median domain onto the adjacent domains. Is there good evidence for that (neuron number; apoptosis..)?

What is the effect of *foxQ2* on the structure of the CB in the adult? How does the phenotype, described only for the late embryo, evolve over time (or do treated animals die?)

Reviewer #3:

This manuscript addresses the contribution of *foxQ2*^+^ neuroblasts and descending neurons to central complex and brain commissure formation in the beetle *Tribolium castaneum*, using genetic labelling and imaging techniques at embryonic and larval stages. First, to identify *foxQ2*^+^ neural progenitors, the authors co-stain for the progenitor marker asense via whole-mount in situ hybridization (wmish) and for *foxQ* via antibody (ab) staining. This reveals four groups of cells present from embryonic into larval stages: anterior-median, anterior-median-posterior, lateral, and ventral progenitor cells. For each group of neural progenitor cells, the authors then determine the transcription factor code at embryonic stages: for example, the conserved regional marker *six3* is co-expressed in all but the ventral group, another conserved mediolateral patterning gene *nk2.1*/scarecrow in the ventral, the brain sensory marker *rx* in the lateral group and additional genes *chx, pax6, fez* in different combinations of all groups. Next, the authors GFP-label the *foxQ2*^+^ neurons with an expression construct using the *foxQ2* promoter. This allows identifying the contributions of *foxQ2*^+^ neurons to the central complex, and to its various substructures at the level of the anterior commissure including the central body. *foxQ2* RNAi knockdown consequentially disturbs commissure structure and abolishes the central body. Furthermore, in vivo imaging reveals morphogenetic movements that shape the larval brain. Finally, the authors present two new imaging lines that they cross with the *foxQ2* line to further elucidate larval brain development and anatomy.

The major value of this study lies in the comprehensive assessment of the role of *foxQ2* to development and anatomy of the insect central complex. The central complex is known to receive and process sensory information from the sensory organs, integrate it with activation state and learning experiences, and provide a first level of motor control output to initiate behavior. A new tailored enhancer trap line allows tracing the *foxQ2* positive neurons through embryonic and larval development and reveals contributions of these neurons to central complex, commissural neuropil and central body substructures such as the central body. This alone is a good step forwards in our understanding of insect brain development and anatomy, addressing a so far neglected area that has recently attracted increased attention by insect neuroanatomists.

From an evolutionary point of view, these data are relevant because the role of *foxQ2* in the formation of apical brain structures appears to be highly conserved in animals but has so far remained enigmatic. (*FoxQ2* is lost in mammals and larval brain development is highly derived in *Drosophila*). Also, the interplay with other conserved apical markers such as *six3, rx, nk2.1, fez* etc. has been barely addressed. *Tribolium* offers a unique opportunity to fill this gap, as it is much more accessible than the fly and shows more ancestral conditions. However, the study falls short of providing new insight into conserved roles of *FoxQ2* in two important points.

- First, much of the imaging is hard to interpret and understanding for a reader not familiar with *Tribolium* embryonic and larval brain anatomy in detail. Explanatory schemes are provided for Figures 1 and 2 but not sufficiently explained and completely missing from the rest of the figures. To an extent that in most cases it remains difficult to appreciate what the authors are referring to. It would be highly desirable to accompany the majority of rows of panels with explanatory drawings that illustrate what the authors want the reader to see. This can be fixed but required substantial effort.

- Second, the neuroanatomical comparative context is not sufficiently clear and only briefly discussed in the last paragraph of the Discussion. How does the central complex exactly relate to the evolution of the apical nervous system? What is the central, or fan-shaped body, and what is its relevance for our understanding of insect brain and apical nervous system evolution? Most important, the authors connect this structure to neuroendocrine centres in other bilaterians. What is this comparison based on, in light of known central complex' motor output? The authors 'sell' their work with the statement that "unexpectedly, our data suggests that the central complex evolved by integrating neural cells from an ancestral anterior neuroendocrine center". This is indeed an intriguing hypothesis, but what cells are they referring to and how does their data support this new hypothesis? As it stands, their hypothesis appears too vague to be convincing.

---

## [Author Response]

Reviewer #1:[…] Overall the manuscript is well written. The figures are excellent and support the narrative. The Discussion is concise and puts the main findings into context. However, the authors seem to directly translate knowledge of developmental processes based on studies in other insects (e.g. grasshopper, Drosophila) to Tribolium (e.g. type I and II neuroblasts, fasciculation patterns; details below) without clarifying that confirmation of similar processes in Tribolium are lacking.

We added respective disclaimers.

(Results) “It should be noted that the *Tc-ase* has been shown to be a marker for NBs in *Tribolium* (Wheeler et al., 2003) but that its expression in INPs or GMCs is assumed by analogy from *Drosophila*.”

(Discussion) “The data suggest that type I and type II neuroblasts are present in *Tribolium* and that both lineages express *Tc-foxQ2*. […] Therefore, we assume that both types of neuroblasts exist in *Tribolium* as well although this remains to be shown.”

(Outlook) “As a basis it will be necessary to systematically compare cell types and markers between *Drosophila* and other insects in order to ascertain that orthologous genes mark homologous cell types.”

Furthermore, the authors discuss what makes the protocerebrum different with regard to the segmental ganglia and state: "The third and most striking role of Tc-foxQ2 in making the protocerebrum different from the segmental ganglia is the requirement for commissure splitting." This statement is not true. In Drosophila, the two segmental commissures of the ventral nerve cord are initially established as one commissure, which is then split into an anterior and posterior commissure by the migration of the midline glia cells. The commissures are subsequently further divided into tracts along the dorso-ventral axis (again by the midline glia).

We are sorry for our imprecise wording – it is not the process of splitting per se but the extend, which we think makes a difference. We have removed the statement from the manuscript.

Finally, the analysis of the brain imaging lines would be more meaningful, if the authors correlated the expression with developing brain centers.

We added an explanation for our focus on the signal that is found in central brain structures because they are relevant for the description of our phenotype. We feel that a comprehensive description of the imaging line would require staining and careful analysis at several stages (embryonic, larval and adult) including 3D reconstructions for visualization. Nevertheless, in the absence of an comprehensive map of neural lineages of the *Tribolium* brain, it would still be difficult to assign these cell groups to certain units.

“Importantly for this work, the line marked the central brain primordium.”

Reviewer #2 (General assessment and major comments (Required)):[…] 1) Results, subsection “Marking of the genetic Tc-FoxQ2 neural lineage by a CRISPR/Cas9 induced enhancer trap” How do the medial and lateral domain described here for antibody and EGFP match the three groups of neural progenitors introduced in subsection “Tc-foxQ2 marks neural progenitor cells with four different molecular identities”? Potential differences in EGFP expression should be mentioned in more detail, referring to the three previously described groups.

Very good point – we had thought a lot about the correlation and had come up with hypotheses – apparently, these thoughts had lost prominence in one of the versions of the manuscript. We have made the respective results more prominent in the Results and added extensive reasoning to the Discussion:

(Results) “In order to assign identified NPCs to this anterior-median-*foxQ2*-cluster we traced back the EGFP signal…”

(Discussion) “Using both the location and *Tc-ase* staining pattern as criteria, we assume that this type I lineage might derive from the *P-fox-l* neuroblast (number 3 in Figure 2I’).”

“…number of *Tc-ase* positive cells in that region decreased over time (*P-fox-am* and *P-fox-amp* in Figure 2). This…”

“The ventrally located group of *Tc-foxQ2* positive NPCs probably contributed to a cell cluster close to the stomodeum, which projected into the circumesophageal commissure but not to the central brain (Figure 5—figure supplement 2). Due to our focus on central brain development, we did not study this group in detail.”

2) Subsection “Three Tc-FoxQ2 positive cell clusters contribute to brain midline structures including the central body”From the description it becomes clear that the foxQ2 Gal4 line remains expressed longer than the gene itself (referring to subsection “Tc-foxQ2 marks neural progenitor cells with four different molecular identities”). The authors should describe this more explicitly. Do the number of lineages that are speculated to form part of the antero-median cluster match the number of progenitors seen earlier (by antibody) in the anteromedial cluster?

Our text was a bit equivocal here. According to our hypothesis, the entire anterior-median group of NPCs belongs to one type II lineage. This explains the pattern of decreasing *ase* positive cells (would be unusual for NBs but expected for INPs). We think that the minor divergence between EGFP signal and *FoxQ2* signal does not influence this reasoning. We now briefly mention this hypothesis in the Results and refer to the Discussion for arguments:

“Based on the number of observed NPCs within the cluster and the number of projections that emerge from it we assume that it represents a type II lineage (see Discussion).”

3) Subsection “Three Tc-FoxQ2 positive cell clusters contribute to brain midline structures including the central body”At the beginning of the section (NS13) the authors propose that the anteromedian clusters contains multiple lineages (second paragraph); at the end, based on glial marker, it sounds like that the cluster corresponds to a single lineage (fifth paragraph). Please clarify.

Indeed, this was inconsistent. We changed according to our hypothesis that we are dealing with one type II lineage (see above).

“Based on the number of observed NPCs within the cluster and the number of projections that emerge from it we assume that it represents a type II lineage (see Discussion for detailed reasoning).”

4) Subsection “Arrest of central brain formation in Tc-foxQ2 RNAi embryos”The terms/concept concerning the "primary commissure" and "splitting of the commissure" need to be clarified. What exactly is defined here as the "primary commissure"? Up until which stage do the authors call the crossing fibers "primary commissure"? Is it a single bundle, or more than one?

We described better the early development as described in the literature

“Development of the protocerebral brain commissures starts with the midline crossing of the primary commissure pioneer neurons. By subsequent joining of additional neurites from the Dm1-4 and other lineages one compact commissure develops. This primary brain commissure later splits and expands into a broad field containing several midline crossing fascicles (Therianos et al., 1995; Williams and Boyan, 2008).”

Concerning the "splitting": It is not clear how the pattern of labeling, with Tubulin antibody or the foxQ2-Gal4, relates to the reality of commissure formation. None of the markers will label all fibers (otherwise, the entire neuropil would be homogenously labeled). There are some fibers/fiber tracts that, at a given time point, will label, whereas others will not (or more weakly). Thus, if one initially sees one single labeled commissural tract, and later two such tracts, separated by a non-labeled "cleft": this could mean (1) that a single commissural bundle has grown in number of its fibers (because neuron of the same class/lineage successively join the bundle), and then a central subset of its fibers (the first born ones?) have lost expression of the marker; or (2) that there was one bundle which does not change, and a second bundle (from a separate class/lineage) has grown alongside, separated by the first by a non-neural cleft (glial sheath?). These distinctions are important when characterizing the RNAi phenotype.

We have to admit that such basic descriptions are still lacking in *Tribolium*. However, we assume that acTubulin marks most fibres at that stage. Further, this process has been extensively studied in the grasshopper such that we can assume with quite some confidence that the major process is the splitting of an initially compact primary commissure. See description above, which is taken mainly from several Boyan papers.

It should also be addressed why the "strong" phenotype resulted in such a massive loss of fiber tracts/neuropil. Was the number of cell bodies (e.g., volume of cortex) equally reduced? Would that fit with the fairly restricted expression of foxQ2 in a few lineages?

Correct, the cell number was decreased, at least those marked by two of our imaging lines (analyzed in the following section). Some reduced cell clusters were of anterior median origin close to the *foxQ2* cells but other affected cell clusters were outside of that domain. We refrained from speculating about the possible mechanisms due to lack of additional evidence.

We added a reference to respective results of the subsequent section:

“Indeed, the number of neurons marked in two imaging lines was reduced (see below)”.

5) Subsections “Analyses in novel brain imaging lines reveals a role for Tc-foxQ2 for different lineages” and “Tc-foxQ2 function is required for survival of neural cells”As indicated above, the authors should address the effect of foxQ2 RNAi on neural lineages in generally more explicitly. If it is true that lineages outside the foxQ2-expressing domain are affected, this would imply some inductive effect of the anterior-median domain onto the adjacent domains. Is there good evidence for that (neuron number; apoptosis..)?What is the effect of foxQ2 on the structure of the CB in the adult? How does the phenotype, described only for the late embryo, evolve over time (or do treated animals die?)

The reviewer is correct in that these are interesting points that we have not been able to address in this paper. We also agree that such an inductive effect would have to be inferred. However, for sake of focus we remained with embryonic development in this study and did not try to find the potentially complex interplay between *foxQ2* knock-down and effects to cells outside the expression domain. These are questions that should be addressed in the future.

Reviewer #3:[…] The study falls short of providing new insight into conserved roles of FoxQ2 in two important points.- First, much of the imaging is hard to interpret and understanding for a reader not familiar with Tribolium embryonic and larval brain anatomy in detail. Explanatory schemes are provided for Figures 1 and 2 but not sufficiently explained and completely missing from the rest of the figures. To an extent that in most cases it remains difficult to appreciate what the authors are referring to. It would be highly desirable to accompany the majority of rows of panels with explanatory drawings that illustrate what the authors want the reader to see. This can be fixed but required substantial effort.

Thanks for this important remark. We hope that we do now provide sufficient explanatory schemes. Specifically, we added more explanation to the schemes in Figure 1 and added a reference for corresponding and directly comparable schemes from *Drosophila* embryos.

Further, we added schemes to relate the fluorescent signal to the embryonic head in Figure 3 (The other figures show embryos of the same stages). Finally, we added schemes to visualize the embryonic movements shown in Figure 6 and amended the respective legends.

- Second, the neuroanatomical comparative context is not sufficiently clear and only briefly discussed in the last paragraph of the Discussion. How does the central complex exactly relate to the evolution of the apical nervous system? What is the central, or fan-shaped body, and what is its relevance for our understanding of insect brain and apical nervous system evolution?

We have expanded the Discussion in the light of the chimeric origin of the brain and speculate about the relationship of apical brain and CX. However, I fear that the data that we contribute with this work do not give crucial additional insights into these very interesting and general questions with respect to CNS evolution. Therefore, we refrained from more speculative inferences.

Most important, the authors connect this structure to neuroendocrine centres in other bilaterians. What is this comparison based on, in light of known central complex' motor output? The authors 'sell' their work with the statement that "unexpectedly, our data suggests that the central complex evolved by integrating neural cells from an ancestral anterior neuroendocrine center". This is indeed an intriguing hypothesis, but what cells are they referring to and how does their data support this new hypothesis? As it stands, their hypothesis appears too vague to be convincing.

We tried to put our idea more clearly in the Discussion. Mainly we discuss two alternative options: chimeric brain hypothesis and de novo evolution of novel functions for these cells in CX formation (see above).